# Manganese-driven CoQ deficiency

Jutta Diessl [1], Jens Berndtsson [2], Filomena Broeskamp [1], Lukas Habernig[1], Verena Kohler[1], Carmela Vazquez-Calvo[1,2], Arpita Nandy[3,4,5], Carlotta Peselj[1], Sofia Drobysheva[1], Ludovic Pelosi[6], F.-Nora Vögtle[3,7,8,9], Fabien Pierrel [6], Martin Ott [2,10] & Sabrina Büttner [1] ✉

Overexposure to manganese disrupts cellular energy metabolism across species, but the molecular mechanism underlying manganese toxicity remains enigmatic. Here, we report that excess cellular manganese selectively disrupts coenzyme Q (CoQ) biosynthesis, resulting in failure of mitochondrial bioenergetics. While respiratory chain complexes remain intact, the lack of CoQ as lipophilic electron carrier precludes oxidative phosphorylation and leads to premature cell and organismal death. At a molecular level, manganese overload causes mismetallation and proteolytic degradation of Coq7, a diiron hydroxylase that catalyzes the penultimate step in CoQ biosynthesis. Coq7 overexpression or supplementation with a CoQ headgroup analog that bypasses Coq7 function fully corrects electron transport, thus restoring respiration and viability. We uncover a unique sensitivity of a diiron enzyme to mismetallation and define the molecular mechanism for manganese-induced bioenergetic failure that is conserved across species.

Manganese (Mn) is a trace metal essential for all forms of life and serves as redox-active cofactor for an array of biological reactions[1–5]. To balance supply and demand, cells need to tightly control Mn assimilation, subcellular distribution, mobilization and efflux[5–7]. Several trace metals, including for instance Mn and iron (Fe), compete for shared trafficking routes and binding sites at macromolecular structures, and deficiency in one metal often results in a relative cellular excess of other metals[8,9]. As more than one-third of all enzymes in animals, plants and microbes are metalloproteins and almost exclusively require one specific metal for catalysis[1,2,10], imbalances in metal pools might affect cellular functionality at all levels. For Mn, the window between dietary adequacy and overload is narrow, and widespread industrial use and increased Mn bioavailability due to soil acidification burden the environment and human health[3,6,11,12]. Across the eukaryotic kingdom, overexposure to Mn disrupts cellular energy metabolism, resulting, for instance, in chlorosis in plants and neurodegeneration in humans[3,4,11,13–15].

Levels of cellular and organellar Mn are established and maintained by the activity of various transporters and pumps in all cellular membranes[9,16]. Though the molecular machinery governing Mn uptake and detoxification differs between species, Mn overload seems to compromise cellular functionality via similar mechanisms across phyla, disrupting in particular energy-converting processes. Current research has mostly focused on Mn-induced neurotoxicity, manifesting in symptoms resembling Parkinson's disease[14,15]. Neuronal processes associated with Mn neurotoxicity are manifold and range from dysregulation of glutamate transport, compromised dopaminergic function, and disturbances of cellular calcium homeostasis to oxidative stress, mitochondrial dysfunction, and disrupted energy metabolism[3,5,17–20], but the fundamental principles of Mn toxicity remain obscure.

Here, we demonstrate that the mitochondrial diiron hydroxylase Coq7, which catalyzes an essential step in coenzyme Q (CoQ)

[1]Department of Molecular Biosciences, The Wenner-Gren Institute, Stockholm University, 10691 Stockholm, Sweden. [2]Department of Biochemistry and Biophysics, Stockholm University, 10691 Stockholm, Sweden. [3]Institute of Biochemistry and Molecular Biology, ZBMZ, University of Freiburg, 79104 Freiburg, Germany. [4]Faculty of Biology, University of Freiburg, 79104 Freiburg, Germany. [5]Spemann Graduate School of Biology and Medicine, University of Freiburg, 79104 Freiburg, Germany. [6]Univ. Grenoble Alpes, CNRS, UMR 5525, VetAgro Sup, Grenoble INP, TIMC, 38000 Grenoble, France. [7]CIBSS - Centre for Integrative Biological Signalling Studies, University of Freiburg, 79104 Freiburg, Germany. [8]Center for Molecular Biology of Heidelberg University (ZMBH), DKFZ-ZMBH Alliance, 69120 Heidelberg, Germany. [9]Network Aging Research, Heidelberg University, 69120 Heidelberg, Germany. [10]Department of Medical Biochemistry and Cell Biology, Institute of Biomedicine, University of Gothenburg, 40530 Gothenburg, Sweden. ✉e-mail: sabrina.buettner@su.se

biosynthesis, is the prime target of Mn overload. Genetic inactivation of Mn detoxification mechanisms or exposure to high environmental Mn resulted in overaccumulation of Mn within mitochondria, leading to mismetallation and subsequent inactivation and degradation of Coq7. In consequence, the lack of the electron carrier CoQ disrupts mitochondrial electron transport, causing respiratory deficiency and premature cell and organismal death. We define mismetallation of a diiron hydroxylase as the cause of Mn toxicity and show that a disequilibrium between cellular Fe and Mn pools directly influences the competition of these metal cofactors for compatible protein binding sites.

## Results

### Cellular Mn overload disrupts mitochondrial bioenergetics

To identify a suitable genetic model for chronic cellular Mn overload, we analyzed the accumulation of trace metals in 19 yeast mutants lacking proteins that have been suggested to be involved in Mn homeostasis using Total reflection X-Ray Fluorescence (TXRF) spectrometry. The absence of Pmr1, a phylogenetically conserved $Ca^{2+}$/$Mn^{2+}$ ATPase localized to the secretory pathway and involved in Mn detoxification[21–24], resulted in a massive overaccumulation of Mn (Fig. 1a and Supplementary Table 1) and also a mild accumulation of other divalent metals, highlighting their interconnected biology (Fig. 1b and Supplementary Table 2). Thus, we employed this model to establish the molecular basis of Mn toxicity. Loss of Pmr1 resulted in a modest growth defect of cells fermenting glucose (Fig. 1c). Upon glucose exhaustion and the switch from fermentative to respiratory metabolism, the lack of Pmr1 caused a rapid and progressive increase of cell death (Fig. 1d), suggesting an abrupt failure of cellular homeostasis when respiration becomes crucial for energy supply. Quantitative multiplexed proteomics revealed a clear change of the proteome upon loss of Pmr1 (Fig. 1e and Supplementary Data 1). Besides a reduced abundance of the majority of mitochondrial proteins in cells lacking Pmr1 (Supplementary Fig. 1a, b), supporting a general decrease in mitochondrial mass (Supplementary Fig. 1c), gene ontology (GO)-term enrichment analysis identified three distinct processes that were most prominently deregulated: cation transmembrane transport, respiratory electron transport chain (ETC) and tricarboxylic acid (TCA) cycle (Fig. 1f). In line, absence of Pmr1 resulted in a severe growth defect on respiratory carbon sources (Fig. 1g), compromised oxygen consumption (Fig. 1h) and reduced mitochondrial transmembrane potential (Fig. 1i). Re-introducing Pmr1 into Δ*pmr1* cells corrected cellular Mn levels (Supplementary Fig. 1d) and restored respiratory growth and cellular survival (Supplementary Fig. 1e, f). Quantification of mitochondrial Mn levels in Δ*pmr1* cells demonstrated that compromised cellular Mn detoxification translated into an overaccumulation of Mn within mitochondria (Supplementary Fig. 1g). Reactive oxygen species, a constant byproduct of oxidative phosphorylation[25] and almost exclusively detectable in mitochondria in wild type cells, were absent in mitochondria from Δ*pmr1* cells but instead produced from extra-mitochondrial sources (Supplementary Fig. 1h). Quantification of Mn levels and oxygen consumption in endogenous point mutants of Pmr1, defective in either Mn transport (Pmr1Q783A) or Ca transport (Pmr1D53A)[26,27], revealed that disruption of mitochondrial bioenergetics was indeed due to loss of Mn transport and subsequent Mn overload, while inactivation of Ca transport had no effect (Fig. 1j, k). Accordingly, environmental Mn exposure faithfully mimicked genetically induced chronic Mn overload, as challenging wild type cells with Mn resulted in a strong cellular Mn overload (Fig. 1l and Supplementary Fig. 1i) and reduced oxygen consumption in a dose-dependent manner (Fig. 1m). We conclude that genetic inactivation of the main Mn detoxification pathway, as well as environmental exposure to high Mn levels, disrupt respiratory metabolism.

### Mitochondrial dysfunction upon Mn overload is a result of coenzyme Q deficiency

To identify genetic modifiers that would restore respiratory function upon Mn overload, we performed a genome-wide screen (Fig. 2a). Δ*pmr1* cells were transformed with a multicopy yeast genomic library[28] and screened for growth on glycerol. Out of roughly $3 \times 10^6$ screened transformants, one plasmid was recovered that could restore respiratory growth and did not code for Pmr1: plasmid pPM6 contained the genes *COQ7* and *IAH1*, coding for enzymes involved in coenzyme Q (CoQ) biosynthesis and isoamyl acetate metabolism, respectively. Individual overexpression revealed that Coq7, but not Iah1, efficiently restored respiratory growth of Δ*pmr1* cells (Fig. 2b). Coq7 is part of an evolutionary conserved multimeric protein complex termed the CoQ synthome[29,30] and functions as a hydroxylase that performs the penultimate step in CoQ biosynthesis (Fig. 2c)[31–34]. The product of the CoQ synthome, the lipid CoQ, carries a redox-active benzoquinone ring attached to a lipophilic polyisoprenoid chain that varies in length across species ($CoQ_6$ in *Saccharomyces cerevisiae*; mostly $CoQ_9$ in Drosophila, and $CoQ_{10}$ in humans) and serves as a crucial electron carrier in the mitochondrial ETC. Thus, we tested whether bioenergetic failure upon Mn overload was due to CoQ deficiency and determined quinone levels in whole cell lipid extracts. Indeed, chronic Mn overload, achieved by genetic ablation of Pmr1 or by selective disruption of Mn transport in the Pmr1Q783A point mutant, resulted in a strong decrease in CoQ and an accumulation of demethoxy-CoQ (DMQ), a precursor of CoQ and the substrate of Coq7 (Fig. 2d–f). Likewise, exposure of wild type cells to external Mn reduced the CoQ levels in a dose-dependent manner and caused an overaccumulation of DMQ (Fig. 2g–i), demonstrating that all upstream steps of the CoQ pathway were functional and that the step catalyzed by Coq7 was selectively impaired.

### Overexpression of Coq7 prevents Mn-driven respiratory failure and restores CoQ levels

To test whether enhanced levels of Coq7 would be sufficient to correct Mn-driven CoQ deficiency, we ectopically overexpressed Coq7. Indeed, increased expression of Coq7 was sufficient to stimulate oxygen consumption in Δ*pmr1* cells and efficiently prevented progressive, age-associated cell death (Fig. 3a, b), despite the persistent overaccumulation of Mn within mitochondria (Fig. 3c). Similarly, mitochondrial dysfunction induced by environmental exposure to excess Mn was prevented by increased Coq7 levels (Fig. 3d). Accordingly, the overexpression of Coq7 restored CoQ levels upon both genetic and dietary induction of Mn overload (Fig. 3e–h). Interestingly, CoQ levels in these cells were even increased above wild type levels, showing that cells devoid of Pmr1 are perfectly equipped to synthesize CoQ when Coq7 amounts are elevated and that a defect in Coq7 function represents the prime cause of Mn toxicity.

### Excess Mn disrupts CoQ-mediated electron transport, while respiratory chain complexes remain intact

Within the mitochondrial electron transport chain, CoQ transfers electrons from NADH or succinate to complex III (CIII). Cytochrome *c* transfers electrons to complex IV (CIV), which subsequently utilizes them to reduce molecular oxygen (Fig. 4a). Despite overaccumulation of Mn within mitochondria isolated from Δ*pmr1* cells (Supplementary Fig. 1g), the formation of the respiratory supercomplexes was unperturbed (Fig. 4b). We assessed the functionality of the electron transport chain complexes in vitro and found that the activity of CIV was slightly increased in Δ*pmr1* mitoplasts (Fig. 4c), while NADH/succinate-independent CIII activity was not affected (Fig. 4d), demonstrating that CIII and CIV are fully functional. Likewise, CIII and CIV remained intact and

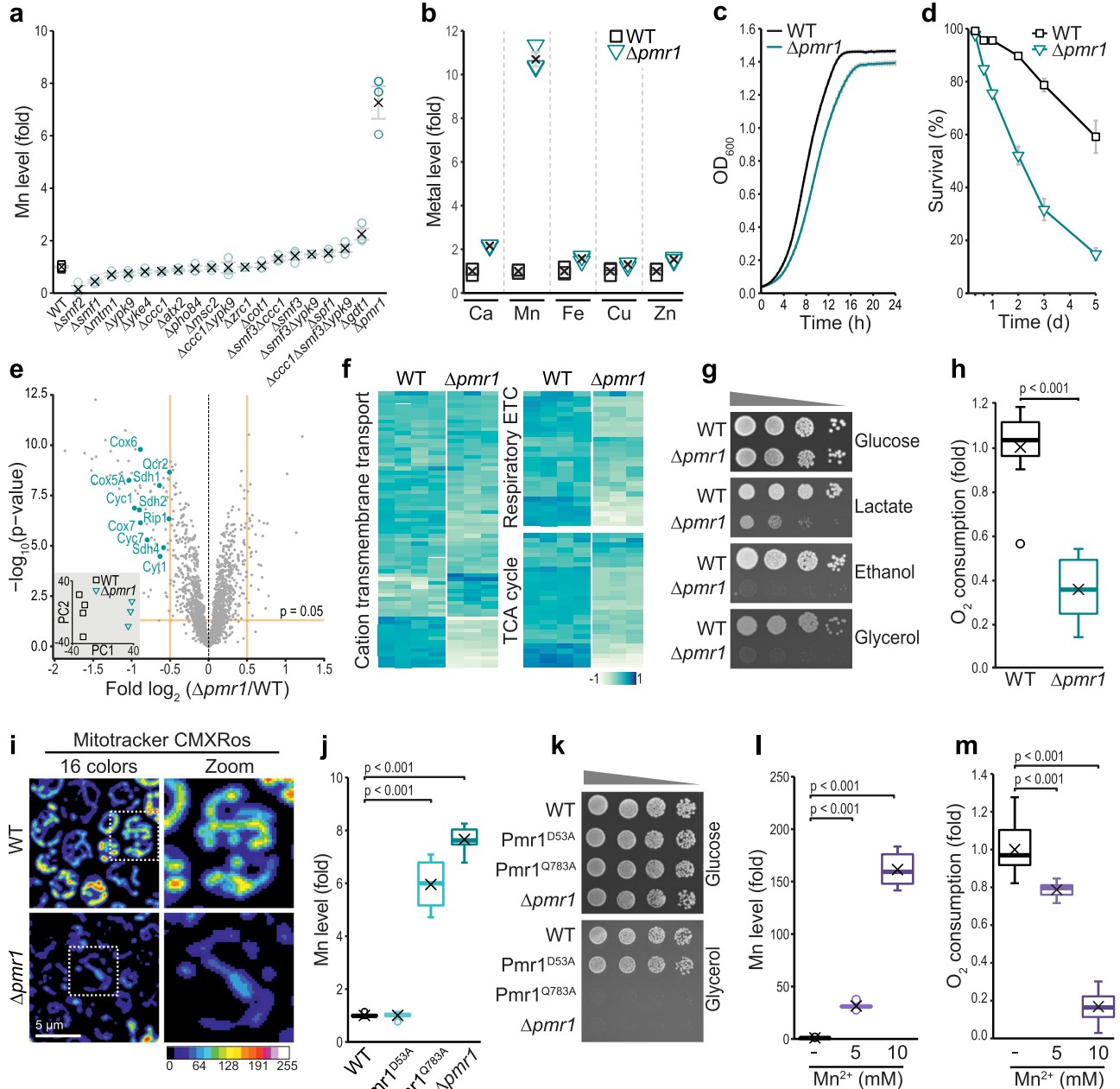

**Fig. 1 | Cellular Mn overload disrupts mitochondrial bioenergetics. a, b** Total cellular Mn (**a**) and metal (**b**) content of wild type (WT) cells and indicated mutants determined via Total reflection X-Ray Fluorescence (TXRF) spectrometry. Cells were collected after 24 h of growth on glucose. Means ± SEM; (**a**) $n = 6$ (WT) or 3 (mutants). (**b**) $n = 3$. **c** Growth kinetics of WT and $\Delta pmr1$ cells grown as described (**a, b**). Means ± SEM, $n = 8$. **d** Survival determined by flow cytometric quantification of propidium iodide staining of WT and $\Delta pmr1$ cells at indicated time points during aging. Means ± SEM, $n = 8$ (8 h and 120 h) or $n = 12$ (16, 24, 48, and 72 h). **e, f** Whole-cell proteomics of WT and $\Delta pmr1$ cells grown as described in (**a, b**). $n = 4$ (WT) or 3 ($\Delta pmr1$). **e** Volcano plot displaying results of differential protein abundance analysis (DeqMS). Significantly less or more abundant proteins of GO term 'Respiratory electron transport chain' are labeled. Inset: Scatterplot of principal component analysis. **f** Heatmaps of identified proteins associated with indicated GO terms.

**g** Spotting of WT and $\Delta pmr1$ cells on glucose, lactate, ethanol, and glycerol. **h** Polarographic determination of oxygen consumption of intact WT and $\Delta pmr1$ cells grown for 24 h on glucose, $n = 11$. **i** Confocal micrographs of cells described in (**h**) stained with Mitotracker CMXRos to visualize mitochondrial transmembrane potential. **j** Total cellular Mn content determined via TXRF of WT, Pmr1[D53A], Pmr1[Q783A], and $\Delta pmr1$ cells after 24 h of growth on glucose media, $n = 8$. **k** Spotting of cells described in (**j**) on glucose and glycerol. **l** Total cellular Mn content determined via TXRF of WT cells grown on glucose and treated with indicated concentrations of MnCl$_2$ for 24 h, $n = 8$. **m** Oxygen consumption of cells described in (**l**), $n = 8$. Box plots (**h, j, l, m**) show mean (x), median (line), first/third quartile (lower/upper bound), minimum/maximum within 1.5-fold IQR (lower/upper whisker), and outliers outside 1.5-fold IQR (circle/o). Details for statistical analysis (Supplementary Table 9) and source data are provided.

functional in mitochondria isolated from wild type cells exposed to excess Mn (Fig. 4e, f). However, we observed a block of electron transfer from NADH or succinate to CIII in mitoplasts from $\Delta pmr1$ cells (Fig. 4g, h) or from wild type cells exposed to high environmental Mn (Fig. 4i). Importantly, an exogenous supply of

decylubiquinone (dCoQ), a soluble CoQ analog, was sufficient to completely restore the electron transfer from both NADH or succinate to CIII (Fig. 4g–i). Similarly, endogenous supply of CoQ, achieved via overexpression of Coq7, efficiently restored electron transfer to CIII (Fig. 4j). In sum, this demonstrates that

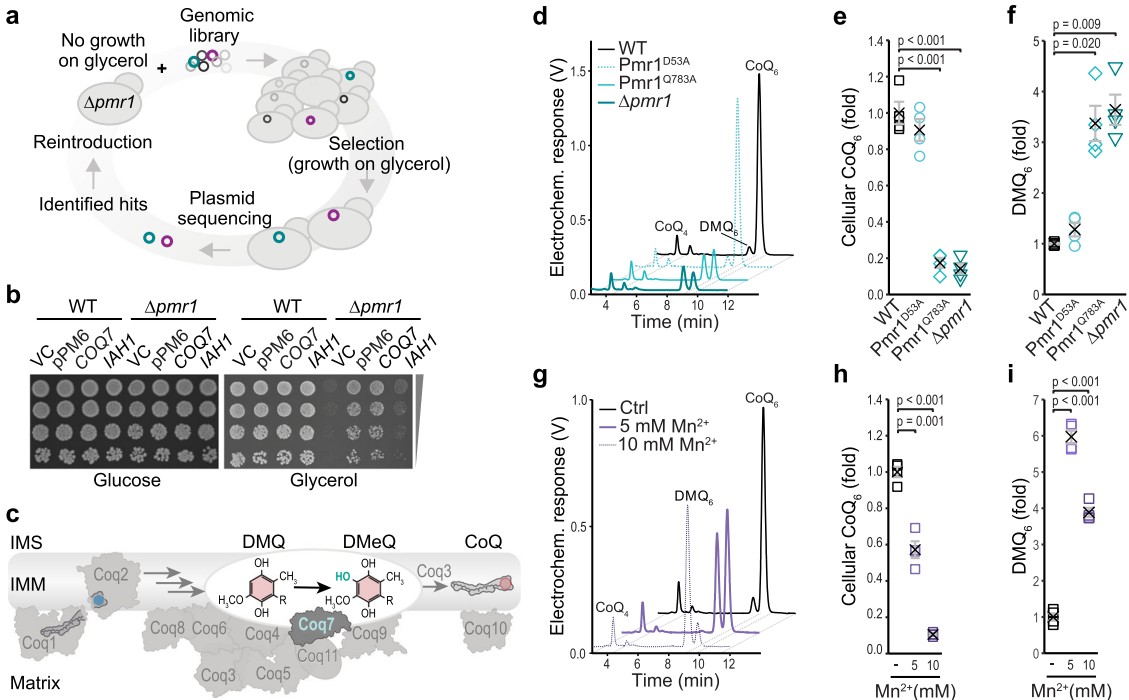

**Fig. 2 | Mitochondrial dysfunction upon Mn overload is a result of CoQ deficiency. a** Schematic illustrating the genome-wide screen to identify suppressors of respiratory deficiency upon Mn overload. **b** Spotting of serial dilutions of Δ*pmr1* cells harboring the vector control (VC), the pPM6 plasmid identified in the screen (containing a library fragment with *COQ7* and *IAH1* genes), or plasmids expressing *COQ7* or *IAH1* under their respective endogenous promoters on plates containing glucose or glycerol as indicated. **c** Representation of coenzyme Q (CoQ) biosynthesis and the CoQ synthome. Coq7 catalyzes the hydroxylation of demethoxy-CoQ (DMQ) to demethyl-CoQ (DMeQ). **d–f** Total cellular CoQ and DMQ content determined via HPLC-ECD of whole cell lipid extracts of WT, Pmr1[D53A], Pmr1[Q783A] and Δ*pmr1* cells. Representative chromatograms from 2.5 OD of cells (**d**) and corresponding quantifications of CoQ (**e**) and DMQ (**f**) are shown. Means ± SEM, n = 4. **g–i** Total cellular CoQ and DMQ content determined via HPLC-ECD of whole cell lipid extracts of WT cells treated with MnCl$_2$ for 24 h. Representative chromatograms from 2.5 OD of cells (**g**) and corresponding quantifications of CoQ (**h**) and DMQ (**i**) are shown. Means ± SEM, n = 4. Details for statistical analysis (Supplementary Table 9) and source data are provided.

mitochondrial Mn overload, achieved by genetic or dietary means, selectively disrupts CoQ biosynthesis, while the complexes of the respiratory chain remain fully functional.

## Mn overload causes mismetallation of the diiron hydroxylase Coq7

Analysis of the steady-state levels of subunits of the CoQ synthome in isolated mitochondria revealed that the levels of Coq6, Coq7, and Coq9 were strongly decreased in the absence of Pmr1, while Coq1 and Coq5 were unaffected (Fig. 5a, b). Interestingly, Coq4, Coq6, Coq7, and Coq9 are part of a submodule of the CoQ synthome, and the loss of one structural component results in loss of the whole module without affecting Coq1 or Coq5[35–37]. Still, overexpression of Coq4 or Coq9, subunits of this module, or of Coq8, suggested to support CoQ synthome assembly, did not support respiratory growth of cells lacking Pmr1 (Fig. 5c and Supplementary Fig. 2a, b). Accordingly, there was no evidence for general deregulation of CoQ metabolism in the proteomic profiling (Fig. 1e) when analyzing the data set for proteins associated with CoQ biosynthesis (Supplementary Fig. 3a) or CoQ uptake and intracellular transport (Supplementary Fig. 3b). In sum, this indicates that mitochondrial Mn overload specifically targets Coq7, which, besides its catalytic function for CoQ biosynthesis, is also crucial for stable module assembly. The reduced abundance of Coq7 caused by cellular Mn overload was neither due to a transcriptional downregulation as determined via qRT-PCR (Supplementary Fig. 4a) nor to defective protein import, as *in organello* import of radiolabeled Coq7 revealed efficient protein translocation into the mitochondrial matrix (Supplementary Fig. 4b). Likewise, the subsequent cleavage of the mitochondrial targeting signal of Coq7 was not affected. Confocal microscopy confirmed the mitochondrial localization of GFP-fusions

of Coq1, Coq7 and Coq9 (Supplementary Fig. 4c). Moreover, Coq7 overexpression in cells lacking Pmr1 did not result in the strong increase of Coq7 protein levels observed in wild type cells (Fig. 5d and Supplementary Fig. 4d). Again, this reduction of steady-state protein levels was not due to changes in transcription (Supplementary Fig. 4e). Even high-level overexpression using alternative promoters, which resulted in a strong accumulation of Coq7 in wild type cells, only slightly increased Coq7 protein levels in cells lacking Pmr1 (Supplementary Fig. 4f, g). To test for premature proteolytic degradation of Coq7 within the mitochondrial matrix, we depleted Pim1, the main mitochondrial protease, using a tetracycline-repressible promoter. Indeed, the inactivation of Pim1 resulted in a strong accumulation of Coq7 in cells lacking Pmr1 (Fig. 5e, f). Collectively, this suggests that Coq7 is unstable and rapidly removed by Pim1 in mitochondria accumulating excess Mn. Still, the slight increase of Coq7 protein achieved via ectopic overexpression was sufficient to restore the steady-state levels of specifically Coq6 and Coq9 (Fig. 5d).

Structurally, the diiron center of Coq7 is coordinated by axial ligands in a four-helix bundle[33], a binding motif shared by other enzymes in the carboxylate-bridged diiron protein family, e.g. mitochondrial alternative oxidase, ribonucleotide reductase, and monooxygenases[33,38]. Insertion of the Coq7 binuclear metal co-factor is thought to be unassisted and hence likely prone to mismetallation. Thus, we tested whether Coq7 inactivation could be due to an erroneous binding of Mn into the site of the diiron center (Fig. 5g) and determined the metallation status of purified Coq7 using TXRF. To enable purification of Coq7 in sufficient quantity from mitochondria of Mn-overaccumulating cells, we not only ectopically expressed FLAG-tagged Coq7 but additionally co-expressed Coq9, known to associate with and stabilize Coq7[34,39,40]. Indeed, in Coq7 purified from Δ*pmr1*

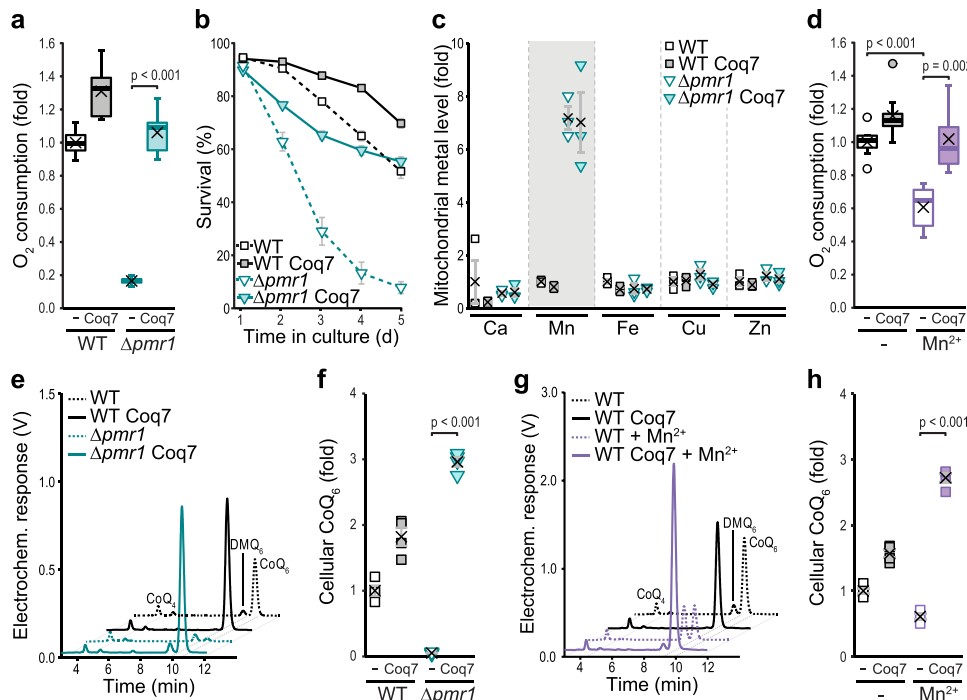

**Fig. 3 | Overexpression of Coq7 prevents Mn-driven respiratory failure and restores CoQ levels. a** Polarographic quantification of oxygen consumption of WT and Δ*pmr1* cells overexpressing Coq7 or harboring the vector control. *n* = 8. **b** Survival of WT and Δ*pmr1* cells overexpressing Coq7 or harboring the vector control determined via flow cytometric evaluation of propidium iodide staining. Means ± SEM, *n* = 8. **c** Metal content of mitochondria isolated from WT and Δ*pmr1* cells overexpressing Coq7 or harboring the vector control determined via TXRF. Means ± SEM, *n* = 3. **d** Oxygen consumption of WT cells overexpressing Coq7 or harboring the vector control grown in glucose media with or without 5 mM MnCl₂ for 24 h. *n* = 8. **e, f** Total cellular CoQ content in lipid extracts of WT and Δ*pmr1* cells

overexpressing Coq7 or harboring the vector control. Representative chromatograms from 2.5 OD of cells (**e**) and corresponding quantifications (**f**) are shown. Means ± SEM, *n* = 4. **g, h** Total cellular CoQ content in lipid extracts of WT cells overexpressing Coq7 or harboring the vector control, treated or not with 5 mM MnCl₂ for 24 h. Representative chromatograms from 2.5 OD of cells (**g**) and corresponding quantifications (**h**) are shown. Means ± SEM, *n* = 4. Box plots (**a**, **d**) show mean (x), median (line), first/third quartile (lower/upper bound), minimum/maximum within 1.5-fold IQR (lower/upper whisker) and outliers outside 1.5-fold IQR (circle/o). Details for statistical analysis (Supplementary Table 9) and source data are provided.

mutants, an increased ratio of Mn to Fe was detected, indicating that at least a part of the protein entraps Mn instead of Fe (Fig. 5h, i). Similarly, exposure to high environmental Mn resulted in an elevated Mn/Fe ratio in purified Coq7 (Fig. 5i). Given that the isolation of metalloproteins in their holo (metal-loaded) form is challenging, as metal ions are frequently lost during the purification process, this finding precludes a precise determination of the portion of Coq7 that incorporates Mn instead of Fe. Thus, we additionally assessed the activity of purified Coq7[31]. Importantly, the in vitro hydroxylation activity of Coq7 purified from Δ*pmr1* mutants or from wild type cells exposed to Mn demonstrated that the increased Mn/Fe ratio resulted in reduced Coq7 activity (Fig. 5j). Metallation is assumed to occur concomitantly with the folding of the nascent polypeptide, offering negligible steric selection[1,2], and binding of the incorrect metal will likely perturb correct folding and thus the activity of Coq7, leading to destabilization and proteolytic removal. In line, mutations in residues of the diiron binding motif inactivate Coq7 across species[41–43]. In yeast, mutation of the Fe binding site as well as insufficient incorporation of Fe into Coq7 via depletion of Fe from mitochondria impairs Coq7 function and causes proteolytic removal[44–46]. Our data demonstrate that excess Mn within the mitochondrial matrix competes with Fe for Coq7 binding sites, which results in mismetallation, inactivation, and degradation of Coq7.

## A CoQ headgroup analog that bypasses Coq7 function restores respiration despite Mn overload

Next, we analyzed whether the CoQ headgroup precursor analog, 2,4-dihydroxybenzoic acid (diHB), would restore mitochondrial function. As diHB already contains the hydroxyl group catalytically added by

Coq7 (Fig. 5k), it selectively corrects defects in CoQ biosynthesis arising from impaired Coq7 function[40,42,43]. However, diHB supplementation cannot rescue *COQ7* deletion mutants due to a complete disruption of the CoQ synthome in the absence of Coq7 as a stabilizing structural component[40]. Indeed, supplementation with this CoQ headgroup analog completely restored respiratory growth of Δ*pmr1* cells but not of Δ*coq7* mutants (Fig. 5l). Thus, the residual amounts of Coq7 in Δ*pmr1* cells can structurally support CoQ synthome assembly to allow sufficient CoQ biosynthesis if supplied with a precursor that bypasses the catalytic activity of Coq7.

## Muscle-specific downregulation of SPoCk compromises CoQ biosynthesis, mitochondrial function and development in Drosophila

Having identified CoQ biosynthesis as the molecular target of Mn toxicity, we tested whether this mechanism is conserved in animals. Hence, we assessed the impact of defective Mn detoxification in the fruit fly *Drosophila melanogaster*. We took advantage of the evolutionary conservation of the secretory pathway Ca²⁺/Mn²⁺-ATPases (SPCAs) and employed transgenic flies carrying a UAS-RNAi directed against SPoCk, the fly homolog of yeast Pmr1 and human SPCAs[47]. Using the muscle-specific *Mef2-Gal4* driver, we selectively depleted SPoCk in muscles, a tissue with high mitochondrial content and energy demand. TXRF-based metal analysis demonstrated that the depletion of SPoCk led to an accumulation of Mn in both larvae and adult flies (Fig. 6a, b). At the cellular level, silencing of SPoCk resulted in compromised mitochondrial function as demonstrated by a decreased mitochondrial transmembrane potential in third instar larval muscle tissue (Fig. 6c, d). Analyses of lipid extracts from larval muscle as well

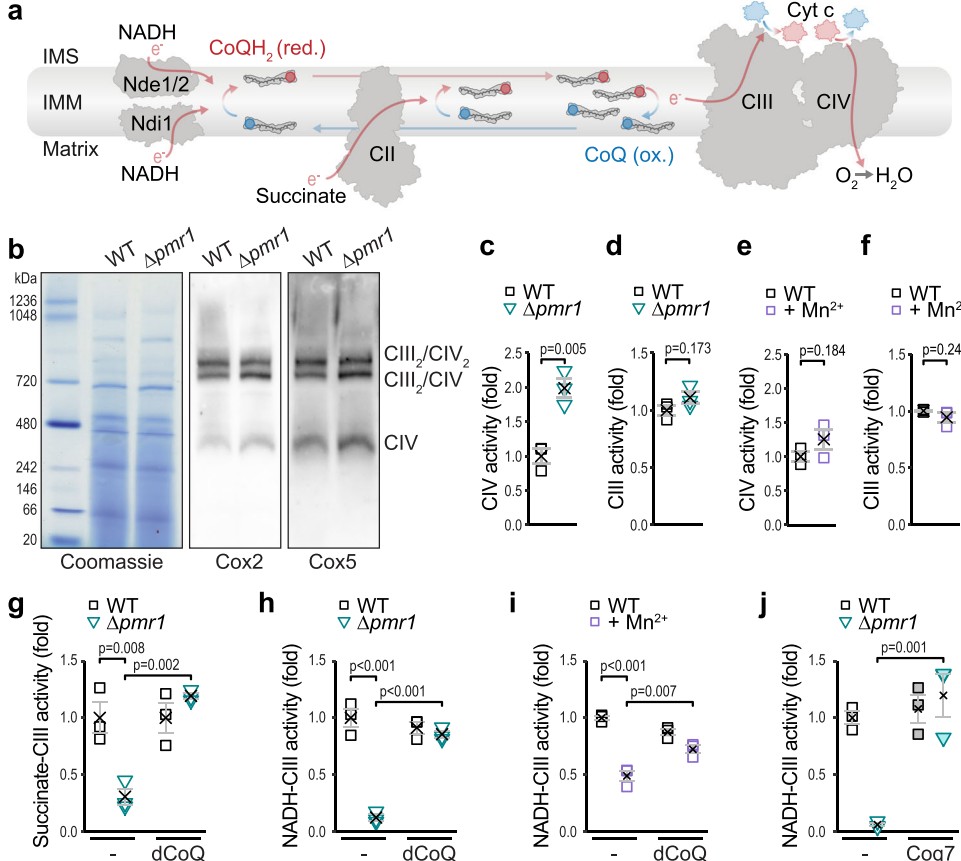

**Fig. 4 | Excess Mn disrupts CoQ-mediated electron transport, while respiratory chain complexes remain intact. a** Schematic depicting the electron flow and the Q cycle in the respiratory chain. IMS = intermembrane space; IMM = inner mitochondrial membrane. **b** Blue-native gel electrophoresis and immunoblots of respiratory supercomplexes in mitochondria isolated from WT and Δ*pmr1* cells. Blots were decorated with antibodies against CIV (Cox2 and Cox5). Representative images of *n* = 3 are shown. **c**–**i** Spectrophotometric analysis of electron flow through the respiratory chain in mitoplasts obtained from mitochondria isolated from the indicated conditions (WT cells with or without 5 mM MnCl₂ or Δ*pmr1*

cells). Cells were grown on glucose for 24 h. CIV activity assessed through oxidation of exogenous reduced cytochrome *c* (cyt *c*) (**c**, **e**), NADH/succinate-independent CIII activity with exogenous decylubiquinol and oxidized cyt *c* (**d**, **f**), succinate-driven CIII activity with and without supplementation of decylubiquinone (dCoQ) (**g**) and NADH-driven CIII activity with and without supplementation of dCoQ (**h**, **i**) are shown. Means ± SEM, *n* = 3. **j** Spectrophotometric analysis of NADH-driven CIII activity in mitoplasts obtained from WT and Δ*pmr1* cells overexpressing Coq7 or harboring the vector control. Means ± SEM, *n* = 3. Details for statistical analysis (Supplementary Table 9) and source data are provided.

as from thorax muscle tissue of these flies just after eclosion revealed a decrease in CoQ₉, the most abundant CoQ in Drosophila (Fig. 6e, f). Monitoring developmental time from first instar larva to adult fly revealed a developmental delay upon muscle-specific depletion of SPoCk (Fig. 6g, h). Dietary supplementation with either CoQ or diHB efficiently corrected these developmental defects (Fig. 6g, h). Moreover, muscle-targeted RNAi attenuation of SPoCk resulted in premature death of a large percentage of flies just after eclosion, and diHB supplementation during larval development improved survival of eclosed adults (Fig. 6i). In sum, muscle-specific knockdown of SPoCk results in decreased CoQ₉ levels and compromised mitochondrial function. These defects lead to a developmental delay that can be completely restored by feeding with CoQ or the headgroup precursor diHB that bypasses Coq7 function. These data demonstrate that the molecular mechanism by which Mn overload provokes mitochondrial dysfunction is conserved from fungi to animals.

## Discussion

Our study uncovers the evolutionary conserved molecular mechanism of Mn toxicity, revealing that the failure of mitochondrial bioenergetics upon overaccumulation of Mn is caused by protein mismetallation, resulting in a disruption of CoQ biosynthesis (Fig. 7). Besides its critical function as an electron carrier, CoQ serves as an antioxidant present in all membranes and supports cellular processes

like fatty acid and pyrimidine metabolism[48,49]. Thus, insufficient synthesis of this lipid disrupts cellular homeostasis at different levels and contributes to a wide range of human pathologies[50]. CoQ deficiencies are broadly classified into primary deficiencies, caused by mutations in genes directly involved in CoQ biosynthesis, and secondary deficiencies, originating from pharmacological treatments, age-related disorders or even aging per se[50,51]. Our study shows that Mn overload causes a strong, secondary CoQ deficiency, which might be particularly relevant in the brain, an organ with high energy demand that has been shown to accumulate Mn[3,6]. Along this line, exposure to excess environmental Mn results in neurotoxicity with symptoms resembling Parkinson's disease. So far, a potential link between Mn-induced neurotoxicity and CoQ deficiency in humans remains speculative. However, our results unravel a fundamental molecular mechanism for Mn toxicity that is conserved from yeast to fly.

At the molecular level, Mn overload, achieved via genetic or dietary means, selectively targets Coq7 and results in inactivation and premature proteolytic removal, while respiratory chain complexes remain functional. Increasing Coq7 protein levels or supplementation of a CoQ headgroup analog that circumvents the catalytic function of Coq7 is sufficient to correct Mn-induced mitochondrial defects. The diiron center of Coq7 is essential for the function and stability of this hydroxylase. Mutations of the E178 residue, which is part of the diiron binding motif, completely inactivates the human and *C. elegans* Coq7

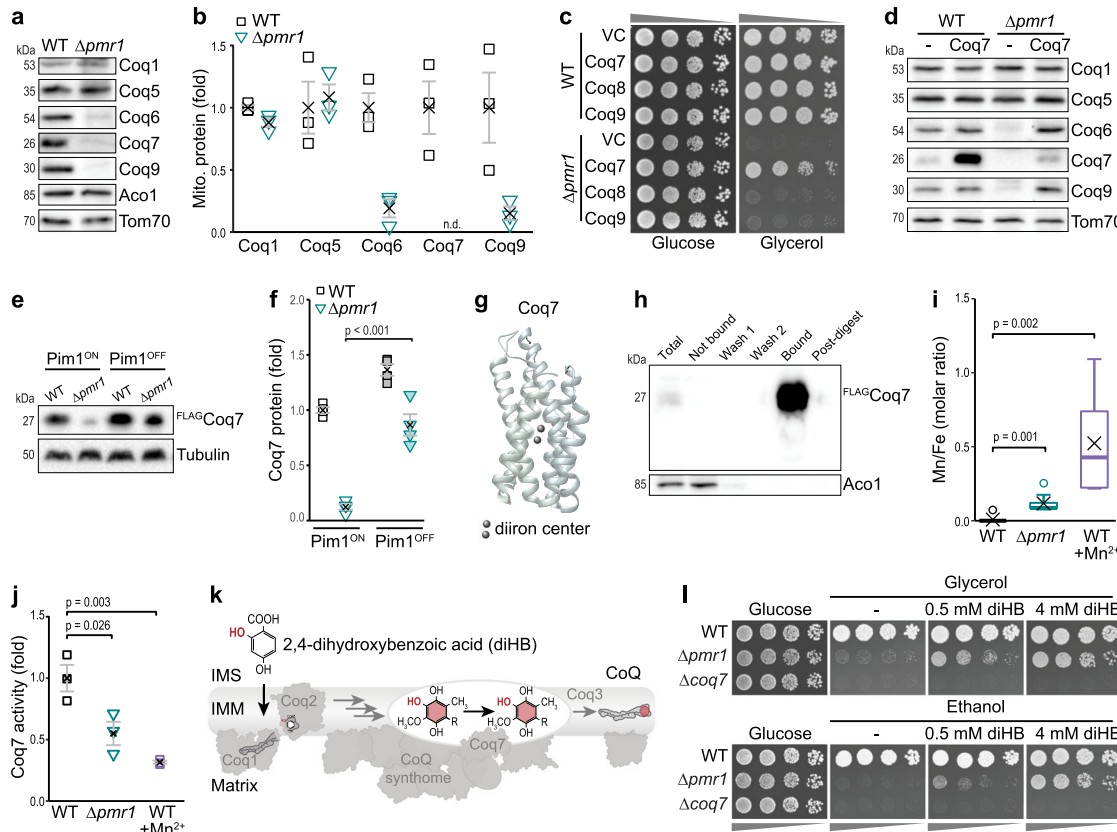

**Fig. 5 | Mn overload causes mismetallation of the diiron hydroxylase Coq7. a, b** Immunoblot analysis of mitochondria isolated from WT and $\Delta pmr1$ cells. Representative blots (**a**) and densitometric quantification for Coq1, Coq5, Coq6, Coq7, and Coq9, normalized to Tom70 and depicted as fold of WT (**b**), are shown. Means ± SEM, $n = 3$. **c** Spotting of WT and $\Delta pmr1$ cells overexpressing Coq7, Coq8 or Coq9 or harboring the vector control (VC) on glucose or glycerol. **d** Representative immunoblot (of $n = 3$) of mitochondria isolated from WT and $\Delta pmr1$ cells overexpressing Coq7 or harboring the vector control. **e, f** Immunoblot analysis of whole cell extracts from WT and $\Delta pmr1$ cells equipped with FLAG-tagged Coq7 and expressing Pim1 under the control of a doxycycline-repressible promoter from its native locus (Pim1$^{ON}$), allowing Pim1 depletion via doxycycline (Pim1$^{OFF}$). Representative immunoblots (**e**) and densitometric quantification of $^{FLAG}$Coq7, normalized to tubulin and depicted as fold of WT Pim1$^{ON}$ (**f**), are shown. Means ± SEM, $n = 4$. **g** Coq7 model predicted with AlphaFold[78] based on amino acid residues 15-233 of Coq7 protein sequence of *S. cerevisiae* (SGDID:S000005651). The two Fe-atoms were placed manually based on[33]. **h** Representative immunoblot (of $n = 3$) of

$^{FLAG}$Coq7 purified with anti-FLAG affinity gel from mitochondria isolated from WT cells overexpressing Coq7 and co-expressing Coq9 to stabilize Coq7 and increase protein yield. **i** TXRF-based determination of the Mn/Fe ratio in Coq7 protein purified from mitochondria isolated from WT and $\Delta pmr1$ cells and from WT cells exposed to 5 mM MnCl$_2$ for 24 h. All cells overexpressed $^{FLAG}$Coq7 and Coq9. Box plot shows mean (x), median (line), first/third quartile (lower/upper bound), minimum/maximum within 1.5-fold IQR (lower/upper whisker), and outliers outside 1.5-fold IQR (circle/o), $n = 9$. **j** In vitro quantification of hydroxylase activity of Coq7 purified from mitochondria from WT and $\Delta pmr1$ cells and from WT cells exposed to 5 mM MnCl$_2$ for 24 h. All cells overexpressed $^{FLAG}$Coq7 and Coq9 as in (**h**, **i**). Means ± SEM, $n = 3$. **k** Schematic representing the bypass of Coq7-catalyzed hydroxylation by 2,4-dihydroxybenzoic acid (diHB). **l** Spotting of WT, $\Delta pmr1$, and $\Delta coq7$ cells on glucose, glycerol, and ethanol plates supplemented with diHB as indicated. Details for statistical analysis (Supplementary Table 9) and source data are provided.

proteins[41,42]. Among the mutations in human Coq7 that result in CoQ deficiency[42,43], the V141E patient allele interferes with the assembly of the diiron center of Coq7[43]. When expressed in mouse embryonic fibroblasts lacking Coq7, this allele only weakly supports CoQ biosynthesis and leads to the accumulation of the precursor DMQ[42]. Similarly, insufficient incorporation of Fe into yeast Coq7 due to mitochondrial Fe depletion or binding site mutation results in Coq7 inactivation and removal[44–46]. Thus, correct metallation of Coq7 during folding of the polypeptide within the mitochondrial matrix is a prerequisite for CoQ biosynthesis.

The biology of Fe and Mn is connected at multiple layers, including shared uptake routes and intracellular storage mechanisms, and Fe deficiencies are often coupled to Mn excess[9,11,12,52]. The divalent ions of Mn and Fe (Mn$^{2+}$ and Fe$^{2+}$) have similar ligand affinities and coordination preferences and thus will compete for often nearly identical protein binding sites[53,54]. For several metalloproteins, the risk of mismetallation is reduced by dedicated metal delivery systems or the use of pre-assembled metal cofactors such as iron-sulfur clusters or heme[1,2]. Incorporation of Fe into these larger cofactors involves

binding to assembly factors before transfer to the ultimate ligand and is highly metal-specific. In contrast, for enzymes containing binuclear metal ion centers such as Coq7, metallation is assumed to be dictated by available metal pools[2,8,55,56]. Still, data on their metal selection and potential mismetallation in vivo is limited[57–60]. We now report that excess Mn disrupts mitochondrial bioenergetics via an erroneous insertion of Mn in the diiron center of Coq7, revealing a unique sensitivity of a membrane-bound diiron carboxylate enzyme towards mismetallation. In contrast to fungi and animals, DMQ hydroxylation in plants and green algae is catalyzed by a flavin-dependent monooxygenase (CoqF)[61–63], likely insensitive towards Mn accumulation. However, membrane-bound diiron carboxylate enzymes in plants include the MME hydroxylase, critical for chlorophyll biosynthesis, as well as the mitochondrial alternative oxidase (AOX) and the plastid terminal oxidase (PTOX), oxidizing ubiquinol and plastochinol, respectively[38]. Thus, members of this family are critical for mitochondrial respiration, chlororespiration, chlorophyll biogenesis, and photosynthesis[38], illustrating that nature uses carboxylate-bridged diiron centers in energy-converting processes across phyla. A

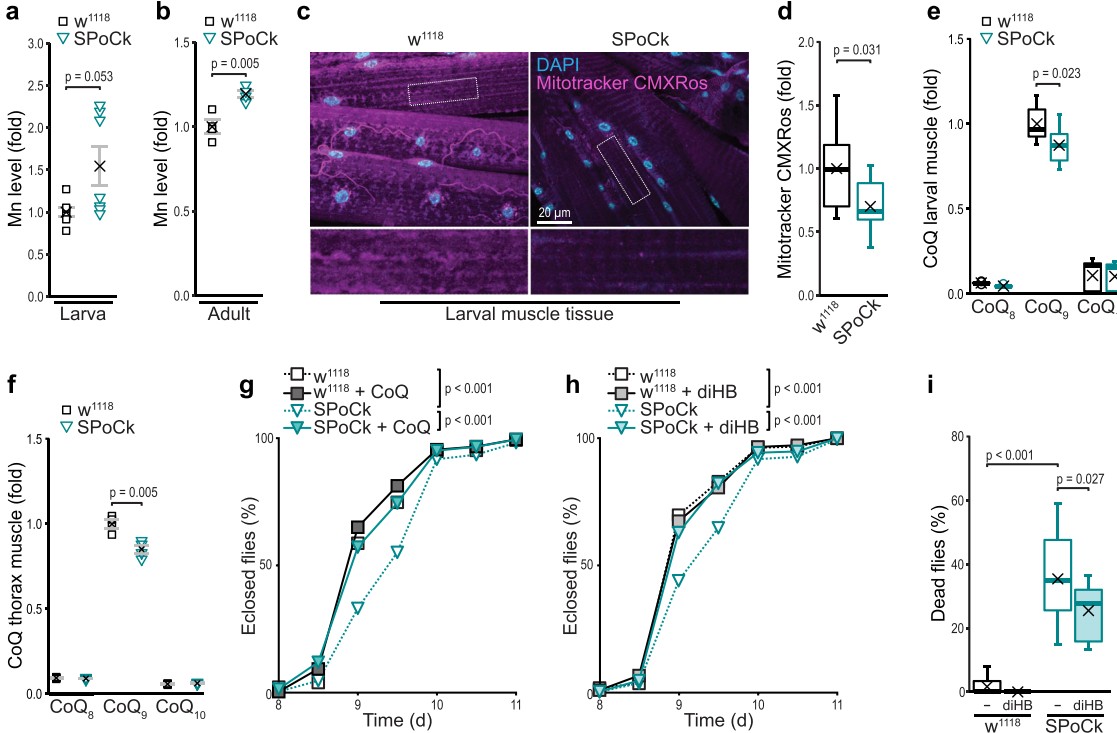

**Fig. 6 | Muscle-specific downregulation of SPoCk compromises CoQ biosynthesis, mitochondrial function and development in Drosophila.** **a, b** TXRF-based determination of Mn levels in third instar larvae (**a**) and adult female flies 1–3 days after eclosion (**b**). Values have been normalized to $w^{1118}$. $n = 7$ (larvae) and $n = 4$ (adults) per genotype (One $n = 10$ larvae or five flies). Means ± SEM. **c, d** Confocal micrographs (**c**) and corresponding quantification of fluorescence intensity (**d**) of $w^{1118}$ (*mef2-Gal4* > ) and SPoCk-RNAi (*mef2-Gal4* > UAS-SPoCk-RNAi) third instar larval muscle tissue stained with Mitotracker CMXRos (magenta) to determine mitochondrial transmembrane potential. Nuclei stained with DAPI (cyan); $n = 10$ ($w^{1118}$) or 9 (SPoCk-RNAi) animals. **e, f** Total CoQ content determined via HPLC-ECD analysis in third instar larval muscle tissue (**e**) and thorax muscle

tissue (**f**) of $w^{1118}$ and SPoCk-RNAi flies 2 days after eclosion (not sex-sorted). Values have been normalized to $CoQ_9$ levels in $w^{1118}$; $n = 9$ (larval muscle) and $n = 4$ (adult thorax) per genotype (One $n = 10$ animals). Means ± SEM. **g, h** Developmental time of $w^{1118}$ and SPoCk-RNAi first instar larvae to adult reared on standard food or supplemented with $CoQ_{10}$ (**g**) or diHB (**h**), $n = 208$-274 animals (**g**), $n = 232$-335 animals (**h**). (**i**) Death of $w^{1118}$ and SPoCk-RNAi adults within 12 h after eclosion reared on standard food or supplemented with diHB, $n > 13$, with 24 flies per n. Box plots (**d, e, i**) show mean (x), median (line), first/third quartile (lower/upper bound), minimum/maximum within 1.5-fold IQR (lower/upper whisker), and outliers outside 1.5-fold IQR (circle/o). Details for statistical analysis (Supplementary Table 9) and source data are provided.

disequilibrium between Fe and Mn pools directly influences the assembly of redox co-factors and thus interferes with essential chemical reactions in life.

## Methods

### Yeast culturing conditions

*S. cerevisiae* cells were grown in synthetic complete medium containing 0.17% yeast nitrogen base, 0.5% $(NH_4)_2SO_4$, 2% glucose, 30 mg/l adenine, 320 mg/l uracil and 30 mg/l of all amino acids (except for 80 mg/l histidine and 200 mg/l leucine). For plasmid selection, media lacking uracil, histidine, or uracil and histidine were used. All components were prepared separately as 10x stocks and mixed after autoclaving (20 min, 121 °C, 210 kPa). Overnight cultures incubated for 16–20 h were used to inoculate cultures to $OD_{600}$ 0.1. Cells were supplemented with 5 or 10 mM $MnCl_2$ at the time of inoculation where indicated. For experiments requiring small amounts of cells (cell death analysis, spotting assay), cells were cultivated in 96-deep well plates at 28 °C, shaking at 999 rpm. For experiments requiring larger amounts of cells (proteomic analysis, TXRF, confocal microscopy, quinone measurements, $O_2$ measurements, qRT-PCR, and immunoblotting), cells were cultivated in 125 ml baffled Erlenmeyer flasks with cellulose stoppers at 28 °C, shaking at 145 rpm.

### Yeast strains and genetics

*S. cerevisiae* strains used in this study were all derived from BY4741 and are listed in Supplementary Table 3. Plasmid transformation was performed as previously described[64]. Plasmids, oligonucleotides,

templates, and restriction enzymes used are listed in Supplementary Tables 4, 5, and 6, respectively. Deletion or endogenous tagging of genes was conducted via homologous recombination following established protocols[65]. Endogenous *PMR1* point mutations and N-terminal insertion of GFP between amino acids 36 and 37 of Coq7 were created following the Delitto perfetto method[66]. pFA6a-kanMX6 was generated by replacing hphNT1 in pFA6a-hphNT1 with a kanMX6 cassette excised from pYM27[65] with BglII and SacI via restriction cloning. For expression of genes under their respective endogenous promoter, the coding region sequence +500 bp upstream of the start codon was amplified. For *COQ4* expression, an additional construct (sPm*COQ4*) with coding sequence +184 bp upstream of the initial ATG was created to avoid potential confounding effects of a *RAV2* gene fragment encoded on the opposite strand (Supplementary Fig. 2b). For ectopic expression of *PMR1* under its own promoter, the coding region sequence +700 bp upstream were amplified. For expression of *COQ7* under control of the tetO-CYC1 or MET25 promoters, the *COQ7* coding region was cloned into pCM190-URA or pUG35-URA, respectively. For the purification of Coq7, a FLAG-tag was inserted between amino acids 36 and 37. Briefly, two overlapping DNA fragments (fragment 1 and fragment 2) containing the FLAG-tag and a spacer region were amplified using pHR81-Coq7 as a template. The two fragments, as well as SacI-/BamHI-linearized empty vector pHR81-URA were transformed into yeast for homologous recombination. All generated constructs were confirmed by sequencing (Eurofins). For construction of the conditional *PIM1* mutant (Pim1$^{ON/OFF}$), the tetO-CYC1 promoter and Tet transactivator domain from pCM190 were restriction cloned into

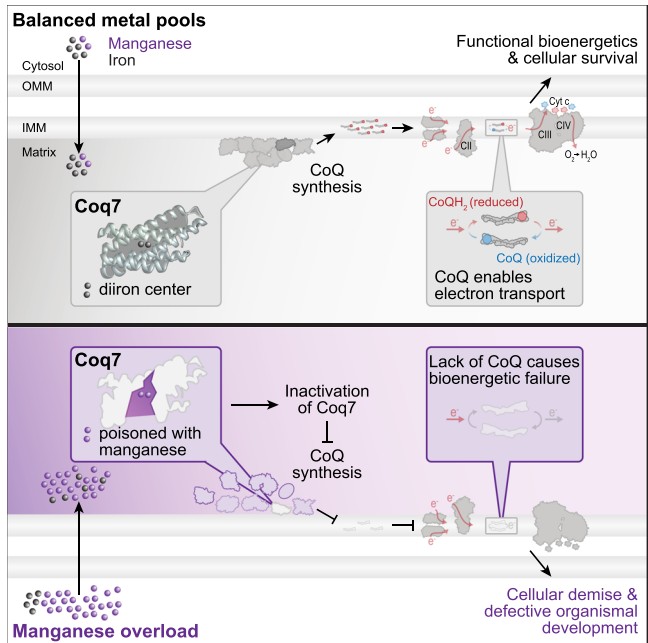

**Fig. 7 | Model of Mn-induced CoQ deficiency, resulting in bioenergetic failure.** Cellular Mn overload, induced by genetic or environmental means, translates into overaccumulation of Mn in mitochondria, which interferes with the correct metallation of the diiron hydroxylase Coq7. In turn, loss of Coq7 compromises CoQ production and precludes CoQ-mediated electron transfer in the electron transport chain. The resulting bioenergetic failure drives cellular and organismal demise.

pFA6a-His3MX6 with EcoRI to create pYM-HIS-N-tetOFF. The whole cassette was PCR-amplified and transformed into yeast, where the resulting promoter exchange enabled inhibition of *PIM1* gene expression upon the addition of 10 µg/ml (final concentration) doxycycline (Sigma, D9891). For marker exchange from *URA3* to *HIS3* to create pHR81-HIS and pHR81-HIS-COQ9 via homologous recombination, *HIS3* was amplified from BY4741 genomic DNA, the pHR81-URA and pHR81-URA-COQ9 plasmids were linearized with HindIII and the fragments transformed into yeast.

### Total reflection X-Ray Fluorescence (TXRF) spectrometry

For whole-cell multi-element analysis, 6 $OD_{600}$ cells were harvested by centrifugation, snap-frozen in liquid nitrogen, and stored at −20 °C until further processing. Cells supplemented with $MnCl_2$, and respective controls were additionally washed with 600 µl 10 mM EDTA. Frozen cell pellets were resuspended in 100 µl 1% Triton X-100 at room temperature (RT), mixed 1:1 with gallium standard (2 mg/l), 10 µl of the sample were transferred to TXRF quartz glass carriers, and dried on a hot plate. For Drosophila samples, whole third instar larvae (one $n = 10$ animals) or whole flies 2 days after eclosion (one $n = 5$ animals) were collected and digested in 70% nitric acid, including 1 ng/µl Ga standard for 48 h. 3 × 5 µl samples were transferred to TXRF quartz glass carriers and dried on a hot plate. Data collection was carried out for 1000 s on an S2 PICOFOX spectrometer (Bruker Nano GmbH, Germany) equipped with a molybdenum excitation source (50 kV/600 µA, max. 50 W (max. 50 kV, max. 1 mA)) and XFlash 430 Picofox detector (Be/30 mm[2]; measured energy resolution 132 eV (Kα-Mn)). Elements were assigned manually and spectra were quantified in the software Spectra v7.8.2. To ensure instrumental precision, gain correction was performed daily with 1 µg As on quartz carrier (supplied by the manufacturer). To ensure the accuracy of measurements, gallium (Ga) was added to each sample in appropriate amounts as internal standard. Sensitivity was tested with 1 ng Ni on quartz carrier (measured value sensitivity: 46

counts $ng^{-1} mA^{-1} s^{-1}$; measured value (lowest limit of detection): 0.104 pg). 3 sample carriers with 1 µl Kraft 13 were used for quantification test, and all measured values (Ti, V, Cr, Mn, Fe, Co, (Ni) Cu, Zn, Rb) were within target values (Ni was set as internal standard). Instrument specifications for the S2 PICOFOX are listed in Supplementary Table 7.

### Analysis of cell growth

Growth kinetics were monitored in 300 µl of culture inoculated to an initial $OD_{600}$ 0.1 at 28 °C in Honeycomb plates on a BioscreenC (Oy Growth Curves Ab Ltd) with continuous shaking, stopping 5 s before measurement. For drop dilution assays, 1 $OD_{600}$ of cells was harvested, and 1:10 serial dilutions were spotted on agar plates containing glucose (4%), lactic acid (3%, pH 5), glycerol (3%), or ethanol (3%) as carbon source. Agar plates contained 0.5 or 4 mM diHB (Merck, D109401-5G) where indicated.

### Quantitative multiplexed proteomics

60 $OD_{600}$ of WT and Δ*pmr1* cells grown for 24 h were harvested by centrifugation and washed with phosphate-buffered saline (PBS) (25 mM potassium phosphate, 0.9% NaCl; pH 7.2). Cell pellets were frozen in liquid nitrogen and stored at −80 °C. Pellets were lysed by 4% SDS lysis buffer and bead milling (Precellys lysing kit VK05). Briefly, beads were combined with 1 ml of re-suspended cells and shaken 8 times (30 hz, 4 min) with 1 min break. Proteins were digested and prepared for mass spectrometry analysis using a modified version of the SP3 protein clean-up and digestion protocol[67]. Peptides were labeled with TMT10plex reagent according to the manufacturer's protocol (Thermo Scientific). Each sample was separated using a Thermo Scientific Dionex nano LC-system in a 4 hr 6-40% ACN gradient coupled to Thermo Scientific High Field QExactive. The software Proteome Discoverer vs. 1.4, including Sequest-Percolator for improved identification, was used to search the *S. cerevisiae* Uniprot database (UP000002311) for protein identification, limited to a false discovery rate of 1%.

### Measurement of O₂ consumption in intact cells

$O_2$ consumption in whole cells was measured in 2 ml of cell culture at 28 °C with a Clarke-type electrode (Rank Brothers Ltd, Cambridge, UK) calibrated with 1x PBS and 30 g/l $Na_2SO_3$ for 5 min each. The decline in $O_2$ concentration in the electrode was monitored over 5 min/sample. Data was collected and analyzed with LabChart v7.3.8. For accurate comparison between strains (WT and Δ*pmr1*), treatments (control and $MnCl_2$), and conditions (vector control and expression construct) the same amount of propidium iodide (PI) negative (live) cells were assayed. Cultures with less PI negative cells were diluted with spent media from the same culture.

### Flow cytometry

Propidium iodide (PI) staining indicative of loss of membrane integrity was used to assess cell death via flow cytometry as previously described[68]. In brief, about $1 × 10^6$ cells were collected in 96-well plates, incubated for 5 min with PI (Sigma 81845; final concentration 500 ng/ml in PBS) and analyzed via flow cytometry using a Guava easyCyte 8HT B/R equipped with a 150 mW 488 nm laser (blue) and a 642 nm laser (red) and the following filters: 488/16 nm (SSC), 525/30 nm (green), and 695/50 nm (red) (Luminex/Merck group). Per sample, 5000 events were recorded. Data was acquired and analyzed using InCyte v3.1, and respective gating strategy is shown in Supplementary Fig. 5a. For determination of mitochondrial mass, 0.2 $OD_{600}$ of cells were pelleted, stained in 250 µl 100 nM MitoTracker Deep Red (Invitrogen™ M22426) in PBS, incubated for 10 min, pelleted and resuspended in PBS and evaluated via flow cytometry using the gating strategy shown in Supplementary Fig. 5b.

## Confocal fluorescence microscopy

1 $OD_{600}$ of cells was harvested by centrifugation and resuspended in PBS-based staining solution with the following dyes and final concentrations: 100 nM Mitotracker CMXRos (Invitrogen™ M7512), 2.5 µg/ml DHE (Sigma D7008) or 100 ng/ml PI (Sigma 81845). Cells were incubated for 10 min in the dark, washed, immobilized on agar slides (3% agar in PBS) and imaged on an LSM700 (Zeiss) confocal microscope with ZEN 2011 black edition v7.0.5.288 software, equipped with a Plan-Apochromat 63x/1.40 oil DIC M27 objective. Excitation/emission wavelengths: 555 nm/566–800 nm for Mitotracker CMXRos and DHE, 488 nm/300–550 nm for GFP, and 488 nm/602–800 nm for PI. Pictures were analyzed and processed with Fiji v1.53q[69]. The micrographs shown are Z-projections obtained by using maximum intensity projection. Images within one experiment were acquired and processed the same way.

## Genomic library screen

The *PMR1* deletion mutant was transformed with the genomic library (pHR81) described in[28] and plated on glucose plates without uracil to allow growth. After 2 days, colonies were washed off with $H_2O$, pooled, and incubated for 30 min (shaking at 140 rpm, 28 °C). Cells were harvested by centrifugation, resuspended in $H_2O$, and frozen as glycerol stocks. *PMR1* deletion mutants containing the genomic library were screened for growth on glycerol plates (3%) without uracil. A total of ca. $3 \times 10^6$ transformants were screened. Library plasmids were isolated, transformed into *E. coli*, extracted and sequenced with primers 5′-GTCTCATCCTTCAATGCTATCA-3′ and 5′-TGTAAAACGACGGCCAGT-3′.

## Isolation of mitochondria and preparation of mitoplasts

Two liters of cell culture per strain and condition were grown for 24 h on synthetic minimal glucose media (without uracil or without uracil and histidine where applicable) and harvested by centrifugation. Mitochondrial isolations were prepared as described previously[70]. Briefly, cells were washed in distilled water, and wet weight (WW) was determined after centrifugation (7 min, 4000 g). After resuspension in 2 ml/(g WW) of MP1 buffer (100 mM Tris, 10 mM DTT, pH unadjusted), the cell suspension was incubated for 10 min (shaking at 170 rpm, 30 °C). Cells were then washed in 1.2 M sorbitol, resuspended in 6.7 ml/(g WW) of MP2 buffer (20 mM potassium phosphate, pH 7.4, 1.2 M sorbitol, 8 mg/(g WW) zymolyase 20 T) and spheroplasted for 35 min (shaking at 170 rpm, 30 °C). Spheroplasts were collected by centrifugation (1800 g, 5 min, 4 °C), resuspended in 6.7 ml/(g WW) of homogenization buffer (10 mM Tris, pH 7.4, 1 mM EDTA, 0.6 M sorbitol and 1 mM phenylmethylsulfonyl fluoride) and homogenized by $2 \times 10$ strokes with a Teflon plunger (Sartorius Stedim Biotech S.A.). Supernatant containing the mitochondrial fraction was harvested after 10 strokes (1800 g, 5 min, 4 °C), and the remaining spheroplasts were resuspended in 6.7 ml/(g WW) and homogenized with 10 more strokes. The two supernatant harvests were pooled and centrifuged once more (1800 g, 5 min, 4 °C) to remove remaining cell debris before pelleting mitochondria by ultracentrifugation (17000 g, 20 min, 4 °C). Intact mitochondria were resuspended in SH-buffer (600 mM sorbitol, 20 mM HEPES-KOH, pH 7.4) and aliquots were stored at −80 °C until further analysis. Mitoplasts were prepared by swelling mitochondria in hypotonic buffer (20 mM HEPES-KOH, pH 7.4) for 30 min (4 °C). Intact mitoplasts were pelleted (25000 g, 20 min, 4 °C), resuspended in swelling buffer, and aliquots were stored at −80 °C.

## Determination of quinone content in yeast and Drosophila

5 $OD_{600}$ of yeast cells were harvested by centrifugation, washed with $H_2O$, pelleted and stored at −80 °C. Quinone extraction was performed as described previously[71], and the dried extracts were resuspended in 100 µl methanol. 50 µl were analyzed for CoQ and demethoxy-$Q_6$ ($DMQ_6$) content by high-performance liquid chromatography-electrochemical detection (HPLC-ECD) as described[71]. For Drosophila samples, inverted third instar larvae were cleaned to reveal muscle tissue (one $n = 10$ animals) or thoraxes were collected from 1–3 days old flies (not sex-sorted) (one $n = 10$ animals) and stored at −80 °C. The samples were homogenized with a Mini-BeadBeater 24 (BioSpec Products, USA) in tubes containing ~300 µl 0.5 mm zirconium beads, 70 µl 0.15 M KCl and 800 µl methanol. Samples were homogenized twice (3200 oscillations/min, 90 s) with 10 min break on ice. 600 µl of petroleum-ether (40–60 °C boiling range) was added to the sample tubes, vortexed (1 min), and the phases were separated by centrifugation (1000 g, 1 min). The p-ether phase containing quinones was collected, the extraction was repeated once, and the two p-ether phases were combined and dried under a nitrogen flow. The dried extracts were resuspended in 100 µl ethanol. 20 µl were analyzed by HPLC-ECD as previously described[72]. $CoQ_8$, $CoQ_9$, and $CoQ_{10}$ present in Drosophila samples were separated and quantified with a standard curve generated with commercial $CoQ_{10}$. Standard curves were generated at each round of analysis by injecting on the HPLC-ECD system various volumes of a stock solution of $CoQ_{10}$ (12.4 µM, as determined from 275 nm absorbance using an extinction coefficient of 15200 $M^{-1}$ $cm^{-1}$ as described[73]), representing quantities of 10–200 pmoles $CoQ_{10}$. The repeatability and the linearity of the calibration curve were excellent.

## SDS-PAGE, Blue-Native PAGE, and immunoblot analysis

SDS-PAGE and immunoblotting of whole cell protein lysates were performed as described[68]. For SDS-PAGE and immunoblotting of isolated mitochondria, samples were thawed on ice, centrifuged (17000 $g$, 15 min, 4 °C), and equal amounts of protein resuspended in SDS sample buffer (50 mM Tris, pH 6.8, 2% SDS, 10% glycerol, 0.01% bromophenol blue, 50 mM DTT added fresh) and boiled for 3 min. Samples were then processed as described[68]. For Blue-native PAGE (BN-PAGE), isolated mitochondria were pelleted (10 min, 16000 g at 4 °C) and lysed in 19 µl BN-PAGE solubilization buffer (50 mM Bis-Tris pH 7.2, 25 mM KCl, 1 mM EDTA, 12% glycerol, 1x Complete Protease Inhibitor cocktail (Roche), 1 mM phenylmethylsulfonyl (PMSF), 1% digitonin) for 20 min at 4 °C. Lysed mitochondria were clarified (8 min, 25000 g at 4 °C) and 1 µl sample additive (5% G-250) was added. Samples were loaded on a 3–12% precast NativePAGE Bis-Tris gel (Invitrogen), ran for about 1/3 of the gel at 150 mV with NativePAGE™ 1x Cathode Buffer and until completion at 250 mV with NativePAGE™ 1x Running Buffer as anode buffer. Next, the gel was either transferred to a PVDF membrane (Bio-Rad; 100 mA, 90 min) or coomassie stained (0.05% BB R-250, 25% isopropanol, 10% acetic acid). For immunoblotting, membranes were decorated with primary mouse antibody raised against FLAG-epitope (1:10000; Sigma-Aldrich, F3165) or primary rabbit antibodies raised against Cox2 (1:500; Neupert lab, Munich, Germany), Cox5 (1:500; Ott lab), Tom70 (1:500; Rapaport lab, Tübingen, Germany), Aco1 (1:5000; Pines lab, Jerusalem, Israel), tubulin (1:10000; Abcam, ab184970) or several Coq proteins (Clarke lab, Los Angeles, USA), including Coq1 (1:200), Coq5 (1:333), Coq6 (1:333), Coq7 (1:1000), and Coq9 (1:400), as well as with the respective peroxidase-conjugated secondary rabbit antibody raised against mouse IgG (1:10000; Sigma-Aldrich, A9044) or secondary goat antibody raised against rabbit IgG (1:10000; Sigma-Aldrich, A0545).

## Spectrophotometry of respiratory chain activity

All respiratory chain activities were assessed spectrophotometrically in mitoplasts at 30 °C as described[74]. Cytochrome *c* (cyt *c*) reduction at 550 nm was used as a readout for NADH/succinate-driven CIII activity with and without exogenous decylubiquinone (dCoQ) and NADH/succinate-independent CIII activity with decylubiquinol (dCoQH₂). Decylubiquinone is a CoQ derivative that can be reduced with lithium borohydride to decylubiquinol. Briefly, NADH/succinate-driven CIII

enzymatic activity was measured in 50 μg mitochondrial proteins in 1 ml 20 mM HEPES buffer (pH 7.4) containing 0.05% oxidized cyt $c$, 1 mg/ml bovine serum albumin (BSA), 240 μM KCN (with 125 μM dCoQ). The reaction was started by addition of 800 μM NADH (final concentration) or 10 mM sodium succinate (final concentration). CIII specificity was confirmed in the same setup via the addition of 0.4 μM antimycin A (final concentration). Similarly, NADH/Succinate-independent CIII activity was measured, except the reaction was started with the addition of dCoQH$_2$ (final concentration 60 μM). The CIII specificity was confirmed in the same setup via the addition of 0.4 μM antimycin A (final concentration). Cyt $c$ oxidation at 550 nm was used as readout for CIV activity. Briefly, CIV activity was measured in 1 ml assay buffer (250 mM sucrose, 20 mM HEPES pH 7.4) containing 0.05% sodium dithionite-reduced cyt $c$, 1 mg/ml BSA, and 0.4 μM antimycin A. The reaction was started by the addition of 50 μg of mitoplasts (10 mg/ml). CIV specificity was confirmed in the same setup via the addition of 240 μM KCN (final concentration).

### Analysis of mRNA levels using quantitative reverse transcription PCR

Total RNA was purified from approximately $2 \times 10^8$ yeast cells with the RiboPure-Yeast™ Kit (AM1926, Invitrogen) according to the supplied manual. For treatment with Turbo DNase I (AM1917, Invitrogen), 20 μl of nucleic acid sample were incubated with 5 μl DNase I buffer and 2 μl of DNase I (8U) for 60 min (37 °C). After 30 min, 8 more units of DNase I were added. The reaction was stopped by adding 0.1 x DNase Inactivation Reagent and vortexing. Samples were incubated for 5 min at RT and centrifuged (3 min, 16 100 g). The supernatant was transferred to a fresh 1.5 ml tube, the concentration of nucleic acids was determined with a Nanodrop (NanoDrop One, Thermo Fisher) and 2 μg were used for reverse transcription with SuperScript II™ (18064014, Invitrogen) according to the manufacturer's instructions. RNaseOUT™ (10777019, Invitrogen) was added to the reaction. DNA contamination was assessed via PCR with primers amplifying *UBC6*. qPCR was performed in a 20 μl reaction with KAPA SYBR® FAST qPCR Master Mix (KM41001, Merck) according to the manufacturer's instructions in a Rotor-Gene Q (QIAGEN) PCR cycler. All primers for qRT-PCR are listed in Supplementary Table 8. Data are represented as fold changes to WT after *COQ* mRNA levels were calculated relative to *UBC6* mRNA levels with the comparative CT method (ΔΔCT)[75].

### *In organello* import of radiolabeled precursor proteins into mitochondria

Radiolabeled precursor proteins were generated using a two-step in vitro transcription and translation reaction system using the Invitrogen™ Ambion mMESSAGE mMACHINE SP6 Transcription Kit and a rabbit reticulocyte lysate system (Promega) with [$^{35}$S] methionine. 40 μg isolated mitochondria and radiolabeled precursor protein were incubated in import buffer (10 mM MOPS-KOH, pH 7.2, 3% w/v BSA, 250 mM sucrose, 5 mM MgCl$_2$, 80 mM KCl, 5 mM KH$_2$PO$_4$, 5 mM methionine) supplemented with 2 mM ATP and 2 mM NADH. Where indicated, 1 μM valinomycin, 20 μM oligomycin and 8 μM antimycin A (AVO mix) were added prior to the import reaction to disrupt mitochondrial transmembrane potential. Import reactions were carried out at 28 °C (Hsp10) or 20 °C (Coq7). AVO mix was added after indicated time to stop the import. Mitochondria were re-isolated by centrifugation and washed with SEM buffer (250 mM sucrose, 1 mM EDTA, 10 mM MOPS-KOH, pH 7.2). Laemmli buffer (2% SDS, 10% glycerol, 0.02% bromophenol blue, 62.5 mM Tris-HCl, pH 6.8, 1% beta-mercaptoethanol) was added to the mitochondrial pellet and heated at 65 °C for 15 min. Hsp10 import samples were analyzed on a 4-12% Nu-PAGE gel, Coq7 import samples on 15% SDS-PAGE. Dried gels were exposed to a phosphor imager screen and radiolabeled proteins were detected by digital autoradiography (PhosphorImager; GE Healthcare).

### Purification of Coq7 from isolated mitochondria and metal content quantification

Mitochondria were isolated from yeast strains co-expressing Coq9 and $^{FLAG}$Coq7 and cultured for 24 h as described above. Mitochondria corresponding to 17 mg of mitochondrial protein were lysed in 1.5 ml lysis buffer (10 mM Tris pH 7.5, 150 mM KCl, 1% digitonin, 1 mM PMSF). After 30 min incubation at 4 °C with rotation, samples were cleared by centrifugation (10 min, 17000 g, 4 °C). ANTI-FLAG® M2 Affinity Gel (A2220, Sigma) was pre-washed three times with 1 mM EDTA and equilibrated with dilution buffer (10 mM Tris pH 7.5, 150 mM KCl, 1% digitonin). 60 μl of pre-washed and equilibrated resin were added to the lysed mitochondrial samples and incubated for 2 h at 4 °C with rotation. Beads were washed three times in ABC buffer (50 mM ammonium bicarbonate, pH 8). Following the last wash, beads were resuspended in 40 μl ABC buffer and an aliquot was taken for Coq7 protein quantification via immunoblotting (bound fraction). Samples were incubated with 1 μg trypsin (T8658, Sigma) for 16 h (37 °C). The digested samples were separated from the beads with spin-columns (BioRad, #7326207), the volume was determined and 5 μl of gallium standard (1 mg/l) were added. The complete volume was applied to TXRF quartz glass carriers in aliquots of 5 μl, which were repeatedly dried on a hot plate to increase sample concentration on the surface. See section on TXRF spectrometry for data collection.

### Coq7 activity measurements

Mitochondrial isolation was performed as described above. 20 mg mitochondria were thawed on ice and lysed for 1 h at 4 °C tumbling (100 mM Tris pH 7.5, 150 mM KCl, 1% digitonin, cOMPLETE mini). After centrifugation (12000 g, 10 min, 4 °C), the supernatant was diluted to 0.2% digitonin and admixed with 60 μl pre-equilibrated and EDTA-treated FLAG beads. The binding step was performed over night at 4 °C tumbling. After 3 washing steps (100 mM Tri pH 7.5, 150 mM KCl, 0.1% digitonin, cOMPLETE mini), elution was performed using 100 μl FLAG peptide solution (150 μg/ml FLAG peptide in washing buffer). The concentration of the eluate was assessed by nanodrop (with Abs 0.1% = 1.038 for Coq7) and all samples were set to 0.01 mg/ml. Coq7 activity was assessed spectrophotometrically by analyzing the decrease in NADH absorbance at 340 nm as previously described[31]. To that end, 135 μl of eluate (0.38 μM) was mixed with 7.5 μl NADH (final concentration of 100 μM) and equilibrated for 5 min at RT. Addition of 7.5 μl 2-Methoxy-5-methyl-1,4-benzoquinone (final concentration 200 μM) marked the start of the enzymatic reaction and absorbance at 340 nm was monitored for 10 min (with a Cary 100 UV-Vis Spectrophotometer from Agilent Technologies and a Q5 High Precision Cell, 10 mm light path from Hellma Analytics). The enzymatic activity was calculated as decrease per time and normalized to untreated wild type samples.

### Drosophila genetics and husbandry

The SPoCk-RNAi (#110379) and Mef2-Gal4 (#27390) lines were obtained from Vienna Drosophila Resource Center and Bloomington Stock Center, respectively. The $w^{1118}$ strain was used as control and for backcrossing of transgenic flies. The muscle-specific SPoCk-RNAi knockdown strain was created by crossing (*mef2-Gal4* > UAS-SPoCk-RNAi). Flies were maintained at 25 °C, 70% humidity and a 12 h/12 h dark/light cycle. All crosses were reared on standard potato mash/molasses medium.

### Developmental assay and death score

Virgins were collected 3-10 days before crossing. Initially, crosses were kept in vials on standard food for 48 h at 25 °C. The following egg collections were performed every 6 h in cages and incubated for 24 h at 25 °C. Plates were supplemented with fresh yeast paste to promote egg-laying. Hatched first instar larvae were sorted and transferred in groups of 24 to vials with 8 ml standard food supplemented with

ethanol, 0.1 mg/ml CoQ$_{10}$ or 6.5 mM diHB (dissolved in ethanol). For developmental assays, the number of eclosed flies was scored every 12 h. Death was scored 12 h after eclosion.

### Larval muscle analysis

Wandering third instar larvae of *mef2-Gal4 > w$^{1118}$* and *mef2-Gal4 > UAS-SPoCk-RNAi* were dissected in Schneider's Drosophila medium, stained for 30 min with 500 nM Mitotracker CMXRos in PBS and washed in PBS. Larva filets were fixed for 30 min in PBS with 4% paraformaldehyde. Fixed tissue was transferred into 500 µl PBS and stained for 30 min with DAPI (1:500). The tissue was washed twice in PBS and mounted with Fluoromount-G™ Mounting Medium (00-4958-02, Thermo Fisher) for image capture using a 63x oil objective on a Zeiss LSM800 confocal microscope using ZEN blue edition v2.3.69.1018 software with identical imaging conditions for all samples. Z-stacks of 30 slices spanning 9.86 µm with a distance of 0.34 µm between slices were used to capture images from the muscle nuclei layer to the top of the muscle surface. A Z-stack projection was used for intensity quantification in Fiji v1.53q. Intensity was measured in 350×350 pixel segments perpendicular to the muscle striations in all animals.

### Data preparation and statistical analysis

Data were analyzed and graphs were generated with R v3.6.2 (base, ggplot2, tidyverse), RStudio v1.2.5[76] and GraphPad Prism v8.0 (developmental assay). Figures were prepared in Adobe Illustrator v26.5. Data sets with n ≥ 8 are presented as box plots with mean (x), median (line), first/third quartile (lower/upper bound), minimum/maximum within 1.5-fold IQR (lower/upper whisker) and outliers outside 1.5-fold IQR (circle/o). Other data are presented as dot plots or line graphs with means and error bars showing standard error of mean (SEM). Principal component, differential protein expression and heatmap analysis were performed on the log2 transformed proteomics data set with `stats::prcomp` function, DeqMS package[77] and gplots package, respectively. For all other data sets where p-values are indicated, normal distribution (Shapiro-Wilk, `stats::shapiro.wilk`) and homogeneity of variances (Levene, `car::leveneTest`) was confirmed, except where noted otherwise (Supplementary Table 9). For comparisons between two groups, an unpaired two-sided Two-sample t-test (`stats::t.test`) was performed for equal variance, a two-sided Welch Two sample t-test for unequal variance (`stats::t.test`) or a two-sided Wilcoxon rank sum test (`stats::wilcox.test`) for data sets with outliers. For comparisons between three or more groups, a One-way ANOVA (`stats::oneway.test`) with Bonferroni post-hoc test was performed (`stats::pairwise.t.test`) for equal variance, a One-way ANOVA not assuming equal variance (`stats::oneway.test`) with Games-Howell post-hoc test (`PMCMRplus::gamesHowellTest`) or a Kruskal-Wallis rank-sum test (`stats::kruskal.test`) with Games-Howell post-hoc test for data sets with outliers or non-normally distributed data with unequal sample sizes. Data of developmental assays were plotted as Kaplan Meier curves and statistics computed with a log-rank test (comparing two groups) and log-rank test with Bonferroni post-hoc test (for multiple comparisons). *P* values in graphs are stated with 3 decimals.

### Reporting summary

Further information on research design is available in the Nature Research Reporting Summary linked to this article.

## Data availability

The mass spectrometry proteomics data have been deposited to the ProteomeXchange Consortium via the PRIDE partner repository with the dataset identifier PXD036952. In addition, the DeqMS analysis of proteomics data is available in Supplementary Data 1. To identify *S.* *cerevisiae* proteins in proteomics analysis, Uniprot proteome UP000002311 was used. Source data are provided with this paper.

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

## Acknowledgements

This work was supported by the Swedish Research Council (2015-05468 and 2019-05249 to S.B., 2018-03694 to M.O.), the Knut and Alice Wallenberg Foundation (2017.009 to S.B. and M.O.), Stiftelsen Olle Engkvist Byggmästare (194-0681 and 207-0527 to S.B.), the Austrian Science Fund FWF (J4342-B21 to V.K.), Deutsche Forschungsgemeinschaft DFG under Germany's Excellence Strategy (CIBSS-EXC-2189, Project ID 390939984 to FNV), SFB1381 (Project ID 403222702 to F.-N.V.), 423813989/GRK2606 (to F.-N.V.) and the Emmy-Noether program (to F.-N.V.), the Medical Research Foundation (DPM2012112553 to F.P.), and the ANR (ANR-19-CE44-0014 to F.P.). Mass spectrometry analysis was performed by the Clinical Proteomics Mass Spectrometry facility, Karolinska Institutet/Karolinska University Hospital/Science for Life Laboratory. We thank Hans Rudolph (University of Stuttgart) for valuable discussions about Pmr1, Catherine Clarke (UCLA) for antibodies against Coq proteins and Martin Högbom (Stockholm University) for providing access to an S2 PICOFOX spectrometer for TXRF.

## Author contributions

Conceptualization, J.D., M.O. and S.B.; Methodology, J.D., J.B., F.B., L.H., V.K., L.P., F.-N.V., F.P., M.O., and S.B.; Investigation, J.D., J.B., F.B., L.H., V.K., C.V.-C., A.N., C.P., S.D., L.P. and F.P.; Formal Analysis, J.D., J.B., C.V.-C., A.N., C.P., S.D., F.B., F.P. and S.B.; Writing – Original Draft, J.D., M.O. and S.B.; Writing – Review and Editing, J.D., F.-N.V., F.P., M.O. and S.B; Funding Acquisition, V.K., F.-N.V., F.P., M.O. and S.B; Supervision, M.O. and S.B.; Visualization: J.D. and S.B. All authors read and approved the final manuscript.

## Funding

## Competing interests

The authors declare no competing interests.
