## [Peer Review File · Nature Communications]

Manganese-driven CoQ deficiencyREVIEWER COMMENTS

Reviewer #1 (Remarks to the Author):

The manuscript describes a novel finding that manganese overexposure selectively disrupts coenzyme Q (CoQ) biosynthesis, resulting in failure of mitochondrial bioenergetics. The manuscript is well written and presented, with a large amount of data supporting the conclusion on mismetallation and loss of a diiron hydroxylase Coq7. This is an important finding which will be of interest to a wide readership.

One minor question to be addressed in revision:

Figure 5g shows Mn/Fe ratio in purified Coq7 protein of delta-pmr1 mutant of only about 0.1. One would expect a higher ratio, up to 1.0, if mismetallation is the main reason for delta-pmr1 phenotype. As a control - is Fe molar content in Coq7 protein from WT cells about 2.0, as would be expected? A discussion on possible reasons for the observed Mn/Fe ratios should be added.

Reviewer #2 (Remarks to the Author):

In this article the authors identify the prime target responsible for the toxicity due to Manganese overload. The toxicity resulting from high Mn is shown to result in the mis-metalation of the polypeptide Coq7, a diiron carboxylate protein that performs the last hydroxylation and penultimate step of CoQ biosynthesis. The experiments are carefully done, and convincing evidence is presented that the decrease in the content of CoQ, is responsible for the toxicity exerted by high Mn, and that this response to Mn overload is conserved from yeast to flies. The implications are quite interesting, as overexposure to Mn has been shown to generate Parkinson's disease-like symptoms in humans.

I believe that the paper could be even better if the authors addressed the following questions:

1. In Figure 1 panel e, a volcano plot is shown that illustrates the differential protein abundance in the pmr1 null cells (which accumulate high Mn content) as compared to WT cells. In this plot a selection of the polypeptide components are labeled. It would be useful to present tabulate all their findings in a Table format. Do other diiron carboxylate proteins show decreased levels? Were the other Coq

polypeptides also identified as being impacted? The authors show later on that Coq4, Coq6, and Coq9 polypeptides were all impacted by high Mn treatment. Were their levels also shown to be impacted in this data set? The data set could also possibly be of interest if the proteins that are significantly induced by high Mn of interest were identified.

2. The effects of high Mn on the CoQ6 content are quite dramatic in the yeast model. While the effect is significant in the *Drosophila* thorax muscle (figure 6 panel c) the decrease observed in the CoQ9 content is much less profound. Indeed, it is surprising that a decrease of just ~10% in the CoQ9 content in thorax muscle would elicit such profound effects on the mitochondrial membrane potential (as measured with Mitotracker CMXRos in Figure 6 panel a). Do the authors have an explanation of why the relatively subtle decrease in CoQ9 content would have such dramatic impact?

3. The authors recovered the Coq7 polypeptide from a yeast over-expression system, and concluded that the Mn:Fe ratio was increased in response to either a lack of *pmr1* or to culture in the presence of high Mn. An activity assay for Coq7 has been published... Did the authors consider testing whether the Coq7 polypeptide recovered from these yeast mitochondrial extracts could be assayed for the hydroxylase activity?

4. Some species do not use a Coq7 type of diiron carboxylate protein to catalyze the last hydroxylation step of CoQ biosynthesis (e.g. some bacteria, plants). Do the authors want to discuss the types of species that would be expected to have CoQ deficiencies in response to high Mn? The authors mention in the introduction that overexposure to Mn disrupts cellular energy metabolism, however, the disruption in plants must be due to different effects than is observed in yeast and flies.

Minor corrections

1. Figure 5 panel i could be more clearly labeled, to indicate that the diHB was added to medium containing glycerol (top panel) or ethanol (bottom panel).

Reviewer #3 (Remarks to the Author):

This manuscript is well written. Major concerns include:

1. In figure 1, with *pmr1* deletion, cellular manganese level increased about 10 folds (1b) and the oxygen consumption decreased over 60% (1h). With 5mM MnCl₂ treatment, cellular manganese increased 100 folds (1i), but the oxygen consumption level only had about 20% decrease.

It is not clear whether the effect of *pmr1* deletion was indeed due to increased cellular manganese in the genetic model. In fact, the present results suggest that cellular alterations other than increased manganese with *pmr1* inactivation likewise contribute to the disrupted mitochondrial bioenergetics.

2. Under manganese overload conditions, the manganese concentration in biological fluid will not reach the millimolar range. The two concentrations, 5mM and 10mM of manganese used in the present study is too high and cannot represent physiological or even pathological conditions.

3. *Pmr1* is a Ca/Mn ATPase in the Golgi apparatus. It remains unclear why *pmr1* deletion in the present study leads to manganese accumulation in mitochondria.

4. In figure 5, it will be useful to compare the stabilities of Coq7 of the WT and *pmr1*-deleted cells.

5. Figure 6 suggests a connection between the fly homologue of *pmr1* and CoQ. However, how does this section establish the relationship between the CoQ deficiency and manganese is unclear.

Reviewer #4 (Remarks to the Author):

Key Results-

The authors use a variety of techniques to demonstrate the effects of excess manganese exposure including impacts on coenzyme Q biosynthesis and subsequent disruptions to respiratory metabolism in *S. cerevisiae* cells.

The authors show that absence of the conserved divalent Ca/Mn P-type ATPase, *Pmr1*, causes significant increases in total manganese. With proteomic data demonstrating decreases in proteins specifically involved in cation transmembrane transport, ETC and TCA. Growth defects, reduced oxygen consumption and mitochondrial transmembrane potential were also demonstrated in $\Delta pmr1$ cells. Respiratory growth of these cells was able to be restored by Coq7 and only this enzyme was impaired during conditions of Mn overload where mismetallation of the diiron center caused its degradation. Furthermore, overexpression of CoQ7 could rescue Mn-driven CoQ deficiency.

Finally, this was demonstrated in vivo by knocking down the fruit fly homolog of *Pmr1* which resulted in decreases mitochondrial function and developmental delay in muscle specific gene knockdown. These

effects were shown to be rescued through dietary intervention with CoQ or an analog of its headgroup precursor.

Significance-

This work employs a number of techniques to identify the effect of increased exogenous manganese exposure and tease out the role of CoQ7 in CoQ biosynthesis in *S. cerevisiae*.

This work provides a platform for which further studies can investigate the holistic effects of disruptions to these pathways in relation to health status during ageing given the proposed link between Mn homeostasis, mitochondrial function and neurotoxicity. Reproducing this work in more complex model organisms will be necessary to test how well these effects translate to higher order organisms with similar transmembrane transport systems.

Data and methodology-

The author's used total reflection x-ray fluorescence (TXRF) to quantify metals which is a well established, highly-sensitive and precise technique for measuring multiple elements in solution. The data set would be strengthened if details on any certified reference materials (CRMs) used as part of the analyses to ensure instrumental accuracy, precision, limits of detection and recovery. Additionally, the raw data generated would be appreciated as all elemental values are reported as fold-differences in comparison to the wild type. Technical specifications of the S2 PICOFOX spectrometer should be supplied as a supplementary table. These suggestions together will improve potential future reproducibility of the data.

The raw proteomic data generated should be included with submission as supplementary in order to be made publicly available.

The raw data generated from the multiplexed proteomics analysis should also be provided as part of the manuscript submission.

It would also be helpful to include the calibration curve (via commercially acquired CoQ10) used to quantify CoQ and DMQ via HPLC-ETD.

Suggested improvements-

The levels of Mn used throughout the study include 5 or 10 mM MnCl₂, given the relative physiological levels of Mn acquired through dietary intake, is this level appropriate for making inferences regarding manganese toxicity that may occur as a result of ageing or acute/chronic exposure to environmental Mn? How did the authors arrive at using this concentration of Mn for their assays and is it feasible that they are well above levels that mammals would encounter in the above-mentioned Mn overload scenarios?

Also, were Mn levels measured in any of the *Drosophila* populations? This would complement the phenotypic effects seen with the growth delays and rescue by dietary CoQ intervention.

Was the complement of the Δ pmr1 strain tested to see if Mn levels were restored to those of wild type?

It would be nice to see direct evidence of CoQ7 mismetallation during conditions of Mn overload, although CoQ7 was overexpressed in wild type then isolated to measure Mn/Fe ratio, how do we know that this increased ratio is not due to non-specific binding of Mn or Fe?

For cells that were prepared for TXRF, it appears that following pelleting via centrifugation the cells are only washed with MilliQ-H₂O, will the rid of the cells of non-specific metal binding sufficiently prior to analysis? Especially in the case of Mn-exposed cells?

Point-by-point reply

First of all, we would like to thank the reviewers for the time and effort spent on our manuscript. We appreciate the constructive and valuable critiques that have been raised and have performed new experiments to carefully address each comment. The new data sets that have been collected support and strengthen our initial conclusions. We believe that following the reviewers' suggestions has substantially improved the quality of our manuscript. Please find below a detailed description of the new results that have been included in response to each comment, the respective new figures and figure panels and the corresponding additions and changes to the manuscript text.

Reviewer #1 (Remarks to the Author)

The manuscript describes a novel finding that manganese overexposure selectively disrupts coenzyme Q (CoQ) biosynthesis, resulting in failure of mitochondrial bioenergetics. The manuscript is well written and presented, with a large amount of data supporting the conclusion on mismetallation and loss of a diiron hydroxylase Coq7. This is an important finding which will be of interest to a wide readership.

We thank the reviewer for these positive comments.

One minor question to be addressed in revision:

Figure 5g shows Mn/Fe ratio in purified Coq7 protein of delta-pmr1 mutant of only about 0.1. One would expect a higher ratio, up to 1.0, if mismetallation is the main reason for delta-pmr1 phenotype. As a control - is Fe molar content in Coq7 protein from WT cells about 2.0, as would be expected? A discussion on possible reasons for the observed Mn/Fe ratios should be added.

This is of course correct. If all Coq7 protein purified from mitochondria of $\Delta pmr1$ cells would be mismetallated, one would expect a higher ratio of Mn/Fe. However, it is in general difficult to isolate metalloproteins in their holo (metal-loaded) form, as metal ions are frequently lost during the process. We cannot exclude that Fe and Mn were partly lost during Coq7 purification and therefore decided to plot the Mn/Fe ratio as a more reliable measure.

Moreover, in presence of high Mn, at least a part of the available Coq7 is still correctly metallated, indicated by e.g. the reduced but not completely absent oxygen consumption in $\Delta pmr1$ cells (still around 40% of wild type cells) as well as the residual amount of CoQ still present in these cells. More importantly, the portion of Coq7 that is mismetallated is rapidly degraded in $\Delta pmr1$ cells. We have now included new data showing that Coq7 is indeed removed via Pim1-mediated proteolysis within mitochondria of $\Delta pmr1$ cells (**new Fig. 5e, f**). For protein purification and subsequent TXRF-based metal measurements, the proteolytic removal of Coq7 from mitochondria of $\Delta pmr1$ cells might lead to an apparently larger amount of correctly metallated Coq7 upon purification. However, we have now performed an *in vitro* assay for Coq7 activity and have directly compared hydroxylase activity of Coq7 purified from mitochondria isolated from (i) wild type cells, (ii) $\Delta pmr1$ cells, and (iii) Mn-treated wild type cells. Importantly, Coq7 purified from Mn-overaccumulating cells (achieved via genetic or dietary means) shows a clearly reduced activity (**new Fig. 5j**). Thus, this new set of data confirms our conclusions, showing that high Mn levels result in Coq7 inactivation. To discuss the limits of the metal measurements in purified Coq7, we have added the following sentence to the respective results section:

*“Given that the isolation of metalloproteins in their holo (metal-loaded) form is challenging, as metal ions are frequently lost during the purification process, this finding precludes a precise determination of the portion of Coq7 that incorporates Mn instead of Fe. Thus, we additionally assessed the activity of purified Coq7³¹. Importantly, the *in vitro* hydroxylation activity of Coq7*

purified from $\Delta pmr1$ mutants or from wild type cells exposed to Mn demonstrated that the increased Mn/Fe ratio resulted in reduced Coq7 activity (Fig. 5j)

Reviewer #2 (Remarks to the Author)

In this article the authors identify the prime target responsible for the toxicity due to Manganese overload. The toxicity resulting from high Mn is shown to result in the mis-metalation of the polypeptide Coq7, a diiron carboxylate protein that performs the last hydroxylation and penultimate step of CoQ biosynthesis. The experiments are carefully done, and convincing evidence is presented that the decrease in the content of CoQ, is responsible for the toxicity exerted by high Mn, and that this response to Mn overload is conserved from yeast to flies. The implications are quite interesting, as overexposure to Mn has been shown to generate Parkinson's disease-like symptoms in humans.

I believe that the paper could be even better if the authors addressed the following questions:

1. In Figure 1 panel e, a volcano plot is shown that illustrates the differential protein abundance in the $pmr1$ null cells (which accumulate high Mn content) as compared to WT cells. In this plot a selection of the polypeptide components are labeled. It would be useful to present tabulate all their findings in a Table format. Do other diiron carboxylate proteins show decreased levels? Were the other Coq polypeptides also identified as being impacted? The authors show later on that Coq4, Coq6, and Coq9 polypeptides were all impacted by high Mn treatment. Were their levels also shown to be impacted in this data set? The data set could also possibly be of interest if the proteins that are significantly induced by high Mn of interest were identified.

We thank the reviewer for this comment. We have now added an excel file as **Supplementary Data 1**, containing a complete list of all proteins detected in the multiplexed proteomics and indicating proteins significantly more or less abundant in the $\Delta pmr1$ mutant. In respect to other proteins of the CoQ synthome that have been detected in the proteomics data set: in total, only 5 Coq proteins were detected in wt and/or $\Delta pmr1$ cells (Coq1, Coq5, Coq8, Coq9 and Coq11), out of which only Coq9 was deregulated (less abundant in $\Delta pmr1$ cells, confirming our findings in Fig. 5a, b). Likewise, additional proteins associated with CoQ biosynthesis or with CoQ transport were not deregulated in cells lacking Pmr1, again suggesting that specifically Coq7 is affected. We have now included an additional **Supplementary Figure 3**, which contains:

- A heatmap of all identified proteins associated with the GO term 'ubiquinone biosynthetic process', showing that out of these proteins, only Coq9 is affected (**new Supplementary Fig. 3a**)
- A volcano plot, displaying the results of differential protein abundance analysis, in which proteins suggested to act in CoQ uptake and intracellular transport have been labelled. This shows that proteins involved in transport of CoQ are not deregulated upon lack of Pmr1 (**new Supplementary Fig. 3b**)

The respective text in the results section has been extended:

“Still, overexpression of Coq4 or Coq9, subunits of this module, or of Coq8, suggested to support CoQ synthome assembly, did not support respiratory growth of cells lacking Pmr1 (Fig. 5c and Supplementary Fig. 2a, b). Accordingly, there was no evidence for a general deregulation of CoQ metabolism in the proteomic profiling (Fig. 1e) when analyzing the data set for proteins associated with CoQ biosynthesis (Supplementary Fig. 3a) or CoQ uptake and intracellular transport (Supplementary Fig. 3b).”

We have identified a few other diiron cluster enzymes in our data set, a few of which are significantly deregulated (the ribonucleotide reductase RNR subunits Rnr2/4, the sphingolipid alpha-hydroxylase Scs7, the sphinganine C4-hydroxylase Sur2), suggesting indeed that Mn overload affects other diiron

cluster enzymes as well. Interestingly, the levels of these proteins were increased, which could hint towards compensatory upregulation upon compromised activity, which has for instance already been suggested for RNR upon treatment with RNR inhibitors. Nonetheless, this is highly speculative at this stage and requires further experimentation. Thus, we prefer to not include a detailed discussion in this respect into the manuscript. Still, the complete proteomics data set is now available as Supplementary Data 1, so interested readers have the possibility to scan for other diiron cluster enzymes of interest.

2. The effects of high Mn on the CoQ6 content are quite dramatic in the yeast model. While the effect is significant in the *Drosophila* thorax muscle (figure 6 panel c) the decrease observed in the CoQ9 content is much less profound. Indeed, it is surprising that a decrease of just ~10% in the CoQ9 content in thorax muscle would elicit such profound effects on the mitochondrial membrane potential (as measured with Mitotracker CMXRos in Figure 6 panel a). Do the authors have an explanation of why the relatively subtle decrease in CoQ9 content would have such dramatic impact?

We have performed additional *Drosophila* experiments and now also include the following data:

- TXRF-based quantification of Mn levels upon depletion of SPoCk in larvae (**new Fig. 6a**)
- TXRF-based quantification of Mn levels upon depletion of SPoCk in adult flies just after eclosion (**new Fig. 6b**)
- Determination of CoQ content in larval muscle tissue (**new Fig. 6e**)

Similar to what we have already shown for adult muscle tissue (thorax), also the CoQ₉ content in larval muscle tissue is reduced by around 10% when SPoCk is depleted. We agree with the referee that this reduction is, though significant, rather mild. However, our measurement of larval and adult muscle CoQ content does not discriminate between mitochondrial and extra-mitochondrial CoQ, and CoQ is of course not only present in mitochondria but also shuttled to all cellular membranes to serve as lipophilic antioxidant. Thus, while our data demonstrates that total CoQ levels are reduced in thorax (and now also in larval muscle, **new Fig. 6e**), it remains to be shown how specifically the mitochondrial CoQ content is affected. Nonetheless, the observed drop in CoQ was sufficient to decrease the mitochondrial transmembrane potential and to cause a developmental delay that could be completely restored by feeding with CoQ or diHB.

As the variation between individual animals was rather high when comparing the mitochondrial transmembrane potential (Mitotracker CMXRos) between wild type and SPoCk-depleted animals in the data set presented in the initial submission (previously Fig. 6b), we have now repeated these experiments and have included additional data points to further strengthen these results (**new Fig. 6d**). In addition, we have now also included TXRF-based quantification of Mn levels upon depletion of SPoCk, showing that this indeed results in Mn overload both in larvae and in adult flies. In line with the subtle drop in CoQ content, also the increase in Mn levels achieved via SPoCk depletion is milder than what we observed upon deletion of Pmr1 in yeast. The results section corresponding to Figure 6 has been adapted according to the new data sets.

3. The authors recovered the Coq7 polypeptide from a yeast over-expression system, and concluded that the Mn:Fe ratio was increased in response to either a lack of pmr1 or to culture in the presence of high Mn. An activity assay for Coq7 has been published... Did the authors consider testing whether the Coq7 polypeptide recovered from these yeast mitochondrial extracts could be assayed for the hydroxylase activity?

Indeed, a Coq7 hydroxylase assay has been established by the Lippard group (PMID: 23445365). We have now set up this *in vitro* assay for Coq7 activity and have directly compared hydroxylase activity of Coq7 purified from mitochondria isolated from (i) wild type cells, (ii) $\Delta pmr1$ cells, and (iii) Mn-treated wild type cells. Importantly, Coq7 purified from Mn-overaccumulating cells (achieved via genetic or dietary means) shows a clearly reduced activity. Thus, this new set of data confirms our conclusions,

showing that high Mn levels result in Coq7 inactivation. The data have been integrated (new Fig. 5j), and the text has been adapted accordingly:

Results:

“... Thus, we additionally assessed the activity of purified Coq7³¹. Importantly, the in vitro hydroxylation activity of Coq7 purified from $\Delta pmr1$ mutants or from wild type cells exposed to Mn demonstrated that the increased Mn/Fe ratio resulted in reduced Coq7 activity (Fig. 5j).”

Discussion:

“At the molecular level, Mn overload, achieved via genetic or dietary means, selectively targets Coq7 and results in inactivation and premature proteolytic removal, while respiratory chain complexes remain functional.”

4. Some species do not use a Coq7 type of diiron carboxylate protein to catalyze the last hydroxylation step of CoQ biosynthesis (e.g. some bacteria, plants). Do the authors want to discuss the types of species that would be expected to have CoQ deficiencies in response to high Mn? The authors mention in the introduction that overexposure to Mn disrupts cellular energy metabolism, however, the disruption in plants must be due to different effects than is observed in yeast and flies.

We have now included an additional paragraph describing CoqF, the flavin-dependent monooxygenase that catalyzes the penultimate step in CoQ synthesis in plants. In addition, we now provide information on other members of the membrane-bound diiron carboxylate enzymes in plants, being involved in chlorophyll synthesis, oxygen-consuming mitochondrial electron transfer chain and oxygen-evolving photosynthetic electron transfer chain. These diiron cluster enzymes might represent potential molecular targets of Mn overload, and reduced activities might contribute to the frequently observed Mn-induced chlorosis in plants. The new section at the end of the discussion reads as follows:

“...revealing a unique sensitivity of a membrane-bound diiron carboxylate enzyme towards mismetallation. In contrast to fungi and animals, DMQ hydroxylation in plants and green algae is catalyzed by a flavin-dependent monooxygenase (CoqF)⁶¹⁻⁶³, likely insensitive towards Mn accumulation. However, membrane-bound diiron carboxylate enzymes in plants include, for instance, the MME hydroxylase, critical for chlorophyll biosynthesis, as well as the mitochondrial alternative oxidase (AOX) and the plastid terminal oxidase (PTOX), oxidizing ubiquinol and plastoquinol, respectively³⁸. Thus, members of this family are critical for mitochondrial respiration, chlororespiration, chlorophyll biogenesis and photosynthesis³⁸, illustrating that nature uses carboxylate-bridged diiron centers in energy-converting processes across phylae. A disequilibrium between Fe and Mn pools directly influences the assembly of these redox co-factors and thus interferes with essential chemical reactions in life.”

Minor corrections

1. Figure 5 panel i could be more clearly labeled, to indicate that the diHB was added to medium containing glycerol (top panel) or ethanol (bottom panel).

We apologize for the lack of clarity. We have now improved the labelling of the spotting assays in former Fig. 5i (now Fig. 5l).

Reviewer #3 (Remarks to the Author)

This manuscript is well written. Major concerns include:

1. In figure 1, with *pmr1* deletion, cellular manganese level increased about 10 folds (1b) and the oxygen consumption decreased over 60% (1h). With 5mM MnCl₂ treatment, cellular manganese increased 100 folds (1i), but the oxygen consumption level only had about 20% decrease.

It is not clear whether the effect of *pmr1* deletion was indeed due to increased cellular manganese in the genetic model. In fact, the present results suggest that cellular alterations other than increased manganese with *pmr1* inactivation likewise contribute to the disrupted mitochondrial bioenergetics.

We have now repeated the quantification of cellular Mn levels in wild type cells upon exposure to high Mn. Reviewer #4 has pointed out that washing Mn-treated cells once in ddH₂O prior to analysis might not be sufficient to get rid of unspecific Mn-binding to the cellular surface. Thus, we performed TXRF-based determination of Mn in cells exposed to 5 mM Mn for 24 h after 1x and 3x wash with ddH₂O as well as after 1x and 3x wash with ddH₂O supplemented with 10 mM EDTA. Indeed, the addition of EDTA decreased Mn-binding to the outside of the cells, resulting in a lower total cellular Mn content upon quantification via TXRF. Still, the exposure to Mn still caused a substantial increase in intracellular Mn (now around 25-fold). Notably, 1x wash in ddH₂O with 10 mM EDTA was sufficient to prevent unspecific Mn-binding to the cells, as 3x wash with EDTA did not reduce Mn levels further. Thus, we have repeated the quantifications of total cellular Mn levels upon exposure of wild type cells to Mn (**new Fig. 1l**), now washing the cells with EDTA prior to analysis. The results of course allow the same conclusion: exposure of wild type cells to Mn results in a prominent increase in intracellular Mn levels, not 100-fold but 25-fold. We have in addition included these control experiments in the Supplementary Information (**new Supplementary Fig. 1i**) and have updated the corresponding methods section.

Regarding the difference in the magnitude of the observed effects, chronic Mn overload (in $\Delta pmr1$ cells) *versus* acute Mn overload (in WT cells upon short-term treatment with high Mn) will of course not result in precisely the same percentages of e.g. the drop in respiration or reduction of CoQ levels. Nonetheless, both chronic, genetically-induced as well as acute, dietary-induced Mn overload result in compromised mitochondrial respiration as well as reduced CoQ levels. Importantly, in both scenarios, simple overexpression of *Coq7* is sufficient to correct these defects. Moreover, in both scenarios, purified *Coq7* shows an increased Mn/Fe ratio and decreased *in vitro* hydroxylation activity (**new Fig. 5j**).

In addition, the lack of *Pmr1* as important Ca/Mn pump in the secretory pathway leads to a more general deregulation of ion homeostasis (as e.g. shown in Fig. 1f, depicting a heatmap for proteins associated with the GO-term “cation transmembrane transport”) and does not only impact Mn homeostasis (please see Fig. 1b, showing metal measurements in $\Delta pmr1$ cells). However, while some other metals do accumulate up to 1.5- to 2-fold in the $\Delta pmr1$ mutant, only Mn is drastically increased (up to 10-fold). Thus, it is feasible that the additional deregulation of other ions contributes to the selective inhibition of *Coq7* upon Mn overaccumulation. Still, our collective data show that Mn overload (achieved by genetic or dietary means) inhibits *Coq7*, thereby disrupting CoQ synthesis and electron transport in the respiratory chain.

2. Under manganese overload conditions, the manganese concentration in biological fluid will not reach the millimolar range. The two concentrations, 5mM and 10mM of manganese used in the present study is too high and cannot represent physiological or even pathological conditions.

It is of course correct that the physiological Mn concentrations for yeast and humans/mammals differ. Still, the Mn concentrations used in this study are well within the physiological range for *S. cerevisiae* (PMID: 19705825 Fig. 1), which can grow from pM to mM external Mn concentrations. While the Mn range used in this study for the yeast experiments might not represent the range found e.g. in biological fluid, also the Mn concentrations used for toxicological studies in mammalian cell culture or on isolated mammalian mitochondria frequently reach 1 mM (cell culture) or 10 mM (isolated mitochondria). Our

data show that in the yeast system, reducing ($\Delta pmr1$ mutant) or overburdening (Mn supplementation) the Mn detoxification capacity of the cell both result in imbalanced metal pools, Mn overaccumulation and inhibition of Coq7. We would argue that our *Drosophila* model, showing that impaired Mn detoxification through SPoCk depletion impairs CoQ biosynthesis, establishes conservation of these results in animals. In line, also the mammalian homolog of Pmr1 (SPCA1) promotes Mn²⁺ detoxification in rat liver cells (PMID: 20981470).

More importantly, in the course of the revision of this manuscript, a new study was uploaded to BioRxiv about 2 weeks ago (<https://www.biorxiv.org/content/10.1101/2022.07.12.499729v1>), showing that in a mouse macrophage cell line, exposure to Mn results in a drop of CoQ and an accumulation of the precursor DMQ, the substrate of Coq7, leading the authors to conclude that Mn interferes with the incorporation of Fe into Coq7. Though this study is limited to the determination of CoQ and DMQ levels upon exposure to Mn, it still shows that our findings are highly relevant for the mammalian system and mechanistically conserved.

3. Pmr1 is a Ca/Mn ATPase in the Golgi apparatus. It remains unclear why *pmr1* deletion in the present study leads to manganese accumulation in mitochondria.

Pmr1 pumps Ca and Mn into the secretory pathway, thereby (i) maintaining the concentrations of these ions in the secretory pathway in the necessary range to support protein folding, processing and glycosylation, and (ii) keeping the cytosolic concentrations of these ions low. The loss of Pmr1 results in overaccumulation of both Mn and Ca in the cytosol and subsequent transport into cellular compartments. While Ca transport into the vacuole is strongly increased in these conditions (e.g. PMID: 26055636, PMID: 10469652), it is less clear how cells handle excess Mn in the cytosol. While also here, cells might increase the transport into the vacuole to reduce the Mn levels in the cytosol, our data indicate that also mitochondria over-accumulate Mn in these conditions (**Fig. 3c and Supplementary Fig. 1g**). This is in line with several previous studies showing that excess cellular Mn results in hyperaccumulation of Mn within mitochondria (e.g. PMID: 10385903; PMID: 1631887; PMID: 17936361), most likely via the mitochondrial Ca uniporter. Though a yeast mitochondrial transporter that imports Mn has not been identified yet, this ion is e.g. crucial for the Mn-dependent superoxide dismutase located within the mitochondrial matrix, indicating that transport routes for Mn into mitochondria do exist, though not yet characterized.

4. In figure 5, it will be useful to compare the stabilities of Coq7 of the WT and *pmr1*-deleted cells.

We thank the reviewer for this suggestion. We now include a new set of data, showing that Coq7 is indeed proteolytically degraded within mitochondria of $\Delta pmr1$ cells (**new Fig. 5e, f**).

We have endogenously replaced the promoter of Pim1, the major protease within the mitochondrial matrix, against a tetracycline-repressible promoter, allowing repression of Pim1 upon doxycycline treatment (and thus avoiding the generation of petite mutants lacking mitochondrial DNA, frequently occurring upon complete deletion of the gene coding for Pim1). While Coq7 protein is hardly detectable in the $\Delta pmr1$ cells, the depletion of Pim1 upon doxycycline treatment results in a prominent accumulation of Coq7 in these cells. This clearly demonstrates that Coq7 is prematurely removed via Pim1-mediated proteolytic degradation in cells accumulating Mn within the mitochondria. The corresponding text has been extended accordingly:

Results:

“To test for premature proteolytic degradation of Coq7 within the mitochondrial matrix, we depleted Pim1, the main mitochondrial protease, using a tetracycline-repressible promoter. Indeed, inactivation of Pim1 resulted in strong accumulation of Coq7 in cells lacking Pmr1 (Fig.

5e, f). Collectively, this suggests that Coq7 is unstable and rapidly removed by Pim1 in mitochondria accumulating excess Mn."

Discussion:

"At the molecular level, Mn overload, achieved via genetic or dietary means, selectively targets Coq7 and results in inactivation and premature proteolytic removal, while respiratory chain complexes remain functional."

5. Figure 6 suggests a connection between the fly homologue of pmr1 and CoQ. However, how does this section establish the relationship between the CoQ deficiency and manganese is unclear.

We have performed additional Drosophila experiments to establish the connection between depletion of SPoCk, manganese overload and CoQ deficiency. The following additional data have been included:

- TXRF-based quantification of Mn levels upon depletion of SPoCk in larvae (**new Fig. 6a**)
- TXRF-based quantification of Mn levels upon depletion of SPoCk in adult flies just after eclosion (**new Fig. 6b**)
- Determination of CoQ content in larval muscle tissue (**new Fig. 6e**)

TXRF-based quantification of Mn levels upon depletion of SPoCk shows that this intervention indeed results in Mn overload both in larvae and in adult flies, connecting the depletion of SPoCk to Mn overaccumulation and to a reduction of CoQ content. Moreover, comparable to what we have already shown for adult muscle tissue (**previously Fig. 6c, now Fig. 6f**), also the CoQ content in larval muscle tissue is reduced when SPoCk is depleted (**new Fig. 6e**). The corresponding text sections have been adapted accordingly.

Reviewer #4 (Remarks to the Author)

Key Results-

The authors use a variety of techniques to demonstrate the effects of excess manganese exposure including impacts on coenzyme Q biosynthesis and subsequent disruptions to respiratory metabolism in *S. cerevisiae* cells. The authors show that absence of the conserved divalent Ca/Mn P-type ATPase, Pmr1, causes significant increases in total manganese. With proteomic data demonstrating decreases in proteins specifically involved in cation transmembrane transport, ETC and TCA. Growth defects, reduced oxygen consumption and mitochondrial transmembrane potential were also demonstrated in Δ pmr1 cells. Respiratory growth of these cells was able to be restored by Coq7 and only this enzyme was impaired during conditions of Mn overload where mismetallation of the diiron center caused its degradation. Furthermore, overexpression of Coq7 could rescue Mn-driven CoQ deficiency. Finally, this was demonstrated in vivo by knocking down the fruit fly homolog of Pmr1 which resulted in decreases mitochondrial function and developmental delay in muscle specific gene knockdown. These effects were shown to be rescued through dietary intervention with CoQ or an analog of its headgroup precursor.

Significance-

This work employs a number of techniques to identify the effect of increased exogenous manganese exposure and tease out the role of Coq7 in CoQ biosynthesis in *S. cerevisiae*. This work provides a platform for which further studies can investigate the holistic effects of disruptions to these pathways in relation to health status during ageing given the proposed link between Mn homeostasis, mitochondrial function and neurotoxicity. Reproducing this work in more complex model organisms

will be necessary to test how well these effects translate to higher order organisms with similar transmembrane transport systems.

Data and methodology-

The author's used total reflection x-ray fluorescence (TXRF) to quantify metals which is a well established, highly-sensitive and precise technique for measuring multiple elements in solution. The data set would be strengthened if details on any certified reference materials (CRMs) used as part of the analyses to ensure instrumental accuracy, precision, limits of detection and recovery.

We have now extended the respective Materials and Methods section with information in respect to instrumental accuracy, limits of detection, calibration and reference material. Moreover, we have also included technical specifications of the S2 PICOFOX spectrometer in **new Supplementary Table 7** (as suggested by the reviewer in the next comment). The following details are given in the Materials and Methods section:

"Frozen cell pellets were resuspended in 100 μ l 1% Triton X-100 at room temperature (RT), mixed 1:1 with gallium standard (2 mg/l), 10 μ l of sample were transferred to TXRF quartz glass carriers and dried on a hot plate. Data collection was carried out for 1000 s on an S2 PICOFOX spectrometer (Bruker Nano GmbH, Germany) equipped with a molybdenum excitation source (50 kV/600 μ A, max. 50 W (max. 50 kV, max. 1 mA)) and XFlash 430 Picofox detector (Be / 30 mm²; measured energy resolution 132 eV (K α -Mn)). Elements were assigned manually and spectra quantified in the software Spectra v7.8.2. To ensure instrumental precision, gain correction was performed daily with 1 μ g As on quartz carrier (supplied by manufacturer). To ensure accuracy of measurements, gallium (Ga) was added to each sample in appropriate amounts as internal standard. Sensitivity was tested with 1 ng Ni on quartz carrier (measured value sensitivity: 46 counts ng⁻¹ mA⁻¹ s⁻¹; measured value (lowest limit of detection): 0.104 pg). 3 sample carriers with 1 μ l Kraft 13 were used for quantification test, and all measured values (Ti, V, Cr, Mn, Fe, Co, (Ni) Cu, Zn, Rb) were within target values (Ni was set as internal standard). Instrument specifications for the S2 PICOFOX are listed in Supplemental Table 7."

Additionally, the raw data generated would be appreciated as all elemental values are reported as fold-differences in comparison to the wild type. Technical specifications of the S2 PICOFOX spectrometer should be supplied as a supplementary table. These suggestions together will improve potential future reproducibility of the data.

As suggested by the reviewer, we have now added the absolute concentrations for the Mn-measurements in the various deletion mutants of genes coding for proteins suggested to be involved in Mn transport or homeostasis (**fold-values still shown in Fig. 1a; corresponding absolute values now listed in Supplementary Table 1**) as well as for the metal measurements comparing wild type and $\Delta pmr1$ cells (**fold-values still shown in Fig. 1b; corresponding absolute values now listed in Supplementary Table 2**). In respect to the technical specifications of the S2 PICOFOX spectrometer, we have now included **new Supplementary Table 7** to provide these details (as mentioned above).

The raw proteomic data generated should be included with submission as supplementary in order to be made publicly available. The raw data generated from the multiplexed proteomics analysis should also be provided as part of the manuscript submission.

We now include the raw and processed data from the multiplexed proteomics as source data and **Supplementary Data 1**, respectively.

It would also be helpful to include the calibration curve (via commercially acquired CoQ10) used to quantify CoQ and DMQ via HPLC-ETD.

We have now included details in respect to the calibration curves, which are routinely generated for each round of analysis, in the corresponding Material and Methods section. The group of Fabien Pierré has quantified CoQ in biological samples by HPLC-ECD for more than 10 years and such quantifications are a routine procedure in his lab. We have added details in the Material and Methods about how the CoQ10 calibration curve is obtained, and we show below typical calibration curves obtained at the beginning (serie 1) and at the end (serie 2) of a sequence of 50 samples. As can be seen below, the linearity is excellent and the coefficients are very close.

The following section has been added to the Materials and Methods section:

“CoQ₈, CoQ₉ and CoQ₁₀ present in Drosophila samples were separated and quantified with a standard curve generated with commercial CoQ₁₀. Standard curves were generated at each round of analysis by injecting on the HPLC-ECD system various volumes of a stock solution of CoQ₁₀ (12.4 μM, as determined from 275 nm absorbance using an extinction coefficient of 15,200 M⁻¹ cm⁻¹ as described ⁷³, representing quantities of 10-200 pmoles CoQ₁₀. The repeatability and the linearity of the calibration curve was excellent.”

Suggested improvements-

The levels of Mn used throughout the study include 5 or 10 mM MnCl₂, given the relative physiological levels of Mn acquired through dietary intake, is this level appropriate for making inferences regarding manganese toxicity that may occur as a result of ageing or acute/chronic exposure to environmental Mn? How did the authors arrive at using this concentration of Mn for their assays and is it feasible that they are well above levels that mammals would encounter in the above-mentioned Mn overload scenarios?

It is of course correct that the physiological Mn concentrations for yeast and humans/mammals differ. Still, the Mn concentrations used in this study are well within the physiological range for *S. cerevisiae* (PMID: 19705825 Fig. 1), which can grow from pM to mM external Mn concentration. While the Mn range used in this study for the yeast experiments might not represent the range found e.g. in biological fluid, also the Mn concentrations used for toxicological studies in mammalian cell culture or on isolated mammalian mitochondria frequently reach 1 mM (cell culture) or 10 mM (isolated mitochondria). Our data show that in the yeast system, reducing ($\Delta pmr1$ mutant) or overburdening (Mn supplementation) the Mn detoxification capacity of the cell both result in imbalanced metal pools, Mn overaccumulation and inhibition of Coq7. We would argue that our *Drosophila* model, showing that impaired Mn detoxification through SPOCK depletion impairs CoQ biosynthesis, establishes conservation of these

results in animals. In line, also the mammalian homolog of Pmr1 (SPCA1) promotes Mn²⁺ detoxification in rat liver cells (PMID: 20981470).

More importantly, in the course of the revision of this manuscript, a new study was uploaded to BioRxiv about 2 weeks ago (<https://www.biorxiv.org/content/10.1101/2022.07.12.499729v1>), showing that in a mouse macrophage cell line, exposure to Mn results in a drop of CoQ and an accumulation of the precursor DMQ, the substrate of Coq7, leading the authors to conclude that Mn interferes with the incorporation of Fe into Coq7. Though this study is limited to the determination of CoQ and DMQ levels upon exposure to Mn, it still shows that our findings are highly relevant for the mammalian system and mechanistically conserved.

Also, were Mn levels measured in any of the *Drosophila* populations? This would complement the phenotypic effects seen with the growth delays and rescue by dietary CoQ intervention.

We have performed additional *Drosophila* experiments to establish the connection between depletion of SPOCK, manganese overload and CoQ deficiency. TXRF-based quantification of Mn levels upon depletion of SPOCK shows that this intervention indeed results in Mn overload both in larvae (**new Fig 6a**) and in adult flies (**new Fig. 6b**). Moreover, comparable to what we have already shown for adult muscle tissue (**previously Fig. 6c, now Fig. 6f**), also the CoQ content in larval muscle tissue is reduced when SPOCK is depleted (**new Fig. 6e**). The corresponding text sections have been adapted accordingly.

Was the complement of the $\Delta pmr1$ strain tested to see if Mn levels were restored to those of wild type?

We apologize for not including these experiments as critical controls. Of course, we have tested for efficient complementation of $\Delta pmr1$ cells by reintroducing Pmr1 and now include these data in Supplementary Figure 1. Ectopic expression of Pmr1 under control of its native promoter efficiently restored the defects of the $\Delta pmr1$ mutant, including (i) Mn levels, determined by TXRF-based metal measurements (**new Supplementary Fig. 1d**), (ii) respiratory growth, determined via serial dilution spotting on the respiratory carbon source glycerol (**new Supplementary Fig. 1e**), and (iii) premature cell death during aging, determined by flow cytometric quantification of propidium iodide staining (**new Supplementary Fig. 1f**). Following text has been included in the results section:

“Re-introducing Pmr1 into $\Delta pmr1$ cells corrected cellular Mn levels (Supplementary Fig. 1d) and restored respiratory growth and cellular survival (Supplementary Fig. 1e, f).”

It would be nice to see direct evidence of CoQ7 mismetallation during conditions of Mn overload, although CoQ7 was overexpressed in wild type then isolated to measure Mn/Fe ratio, how do we know that this increased ratio is not due to non-specific binding of Mn or Fe?

We agree with the reviewer. While all of our data suggests that hyperaccumulation of Mn within mitochondria interferes with the incorporation of Fe into this diiron cluster enzyme, resulting in premature proteolytic degradation (we have now added additional data in this respect in Fig. 5e,f, showing that Mn overload causes Pim1-mediated degradation of Coq7 within mitochondria), we can indeed not exclude that the increased Mn/Fe ratio observed in this scenario is not (at least in part) due to unspecific binding of Mn to the protein. However, it is in general methodologically challenging to isolate metalloproteins in their holo (metal-loaded) form, as metal ions are frequently lost during the purification process, and TXRF-based metal measurements do of course only determine the quantity, not the specific location, of the metals present in purified proteins. However, we have now included *in vitro* enzyme activity assays for Coq7 and have directly compared hydroxylase activity of Coq7 purified from mitochondria isolated from (i) wild type cells, (ii) $\Delta pmr1$ cells, and (iii) Mn-treated wild type cells. Indeed, Coq7 purified from Mn-overaccumulating cells (achieved via genetic or dietary means) shows a clearly reduced activity. Thus, this new set of data confirms that high Mn levels result in Coq7 inactivation. The data has been integrated (**new Fig. 5j**), and the text has been adapted accordingly:

“Given that the isolation of metalloproteins in their holo (metal-loaded) form is challenging, as metal ions are frequently lost during the purification process, this finding precludes a precise determination of the portion of Coq7 that incorporates Mn instead of Fe. Thus, we additionally assessed the activity of purified Coq7³¹. Importantly, the in vitro hydroxylation activity of Coq7 purified from $\Delta pmr1$ mutants or from wild type cells exposed to Mn demonstrated that the increased Mn/Fe ratio resulted in reduced Coq7 activity (Fig. 5j).”

For cells that were prepared for TXRF, it appears that following pelleting via centrifugation the cells are only washed with MilliQ-H₂O, will the rid of the cells of non-specific metal binding sufficiently prior to analysis? Especially in the case of Mn-exposed cells?

We thank the reviewer for this comment. We have performed additional experiments to test whether the simple wash with ddH₂O was sufficient to get rid of Mn binding to the cell wall upon exposure of cells to Mn in the medium. TXRF-based determination of Mn in cells exposed to 5 mM Mn for 24 h was performed after 1x and 3x wash with ddH₂O as well as after 1x and 3x wash with ddH₂O supplemented with 10 mM EDTA. Indeed, the addition of EDTA decreased Mn-binding to the outside of the cells, resulting in a lower total cellular Mn content upon quantification via TXRF. Still, the exposure to Mn still caused a substantial increase in intracellular Mn (around 25-fold). Notably, 1x wash in ddH₂O with 10 mM EDTA was sufficient to prevent unspecific Mn-binding to the cells, as 3x wash with EDTA did not reduce Mn levels further. Thus, we have repeated the quantifications of total cellular Mn levels upon exposure of wild type cells to Mn (**new Fig. 1l**), now washing the cells with EDTA prior to analysis. The results of course allow the same conclusion: exposure of wild type cells to Mn results in a prominent increase in intracellular Mn levels, but only 25-fold and not 100-fold. We have in addition included these control experiments in the Supplementary Information (**new Supplementary Fig. 1i**) and have updated the corresponding methods section.

REVIEWERS' COMMENTS

Reviewer #2 (Remarks to the Author):

The authors have done a commendable job responding to the reviewer's suggestions and queries. The additional experiments are carefully done and the results are convincing.

The authors should review Supplemental Figure 1 panel e. This plate dilution assay doesn't make sense as labeled, because As presented, the WT strain is unable to grow on glycerol if a plasmid expressing Pmr1 is expressed, and the pmr1 null is shown to grow on the glycerol medium whether or not the Pmr1 expressing plasmid is present.

Reviewer #3 (Remarks to the Author):

Previous major concerns have been addressed by the authors.

Reviewer #4 (Remarks to the Author):

Thank you to the authors for their diligent work to address my comments. I believe that all of my comments and concerns have been sufficiently addressed.

Point-by-point reply

REVIEWERS' COMMENTS

Reviewer #2 (Remarks to the Author):

The authors have done a commendable job responding to the reviewer's suggestions and queries. The additional experiments are carefully done and the results are convincing.

The authors should review Supplemental Figure 1 panel e. This plate dilution assay doesn't make sense as labeled, because as presented, the WT strain is unable to grow on glycerol if a plasmid expressing Pmr1 is expressed, and the pmr1 null is shown to grow on the glycerol medium whether or not the Pmr1 expressing plasmid is present.

We apologize for this mis-annotation and thank the reviewer for pointing it out. We have now corrected this mistake.

Reviewer #3 (Remarks to the Author):

Previous major concerns have been addressed by the authors.

Reviewer #4 (Remarks to the Author):

Thank you to the authors for their diligent work to address my comments. I believe that all of my comments and concerns have been sufficiently addressed.